| Editor's Pick | Human Microbiome | Research Article

# First-year dynamics of the anaerobic microbiome and archaeome in infants' oral and gastrointestinal systems

Charlotte J. Neumann,[1] Rokhsareh Mohammadzadeh,[1] Pei Yee Woh,[2,3] Tanja Kobal,[1] Manuela-Raluca Pausan,[1,4] Tejus Shinde,[1] Victoria Haid,[1] Polona Mertelj,[1] Eva-Christine Weiss,[5] Vassiliki Kolovetsiou-Kreiner,[5] Alexander Mahnert,[1] Christina Kumpitsch,[1] Evelyn Jantscher-Krenn,[5,6,7] Christine Moissl-Eichinger[1,7]

**ABSTRACT** Recent research provides new insights into the early establishment of the infant gut microbiome, emphasizing the influence of breastfeeding on the development of gastrointestinal microbiomes. In our study, we longitudinally examined the taxonomic and functional dynamics of the oral and gastrointestinal tract (GIT) microbiomes of healthy infants ($n = 30$) in their first year, focusing on the often-over-looked aspects, the development of archaeal and anaerobic microbiomes. Breastfed (BF) infants exhibit a more defined transitional phase in their oral microbiome compared to non-breastfed (NBF) infants, marked by a decrease in *Streptococcus* and the emergence of anaerobic genera such as *Granulicatella*. This phase, characterized by increased alpha-diversity and significant changes in beta-diversity, occurs earlier in NBF infants (months 1–3) than in BF infants (months 4–6), suggesting that breastfeeding supports later, more defined microbiome maturation. We demonstrated the presence of archaea in the infant oral cavity and GIT microbiome from early infancy, with *Methanobrevibacter* being the predominant genus. Still, transient patterns show that no stable archaeome is formed. The GIT microbiome exhibited gradual development, with BF infants showing increased diversity and complexity between the third and eighth months, marked by anaerobic microbial networks. NBF infants showed complex microbial co-occurrence patterns from the start. These strong differences between BF and NBF infants' GIT microbiomes are less pronounced on functional levels than on taxonomic levels. Overall, the infant microbiome differentiates and stabilizes over the first year, with breastfeeding playing a crucial role in shaping anaerobic microbial networks and overall microbiome maturation.

**IMPORTANCE** The first year of life is a crucial period for establishing a healthy human microbiome. Our study analyses the role of archaea and obligate anaerobes in the development of the human oral and gut microbiome, with a specific focus on the impact of breastfeeding in this process. Our findings demonstrated that the oral and gut microbiomes of breastfed infants undergo distinct phases of increased dynamics within the first year of life. In contrast, the microbiomes of non-breastfed infants are more mature from the first month, leading to a steadier development without distinct transitional phases in the first year. Additionally, we found that archaeal signatures are present in infants under 1 year of age, but they do not form a stable archaeome. In contrast to this, we could track specific bacterial strains transitioning from oral to gut or persisting in the gut over time.

**KEYWORDS** gut microbiome, GIT, oral microbiome, infant development, early life, metagenomics, anaerobes, archaea, strain tracking, source tracking

**Peer Reviewers** Elaine M. Haase, University at Buffalo, Buffalo, New York, USA; Justin Shaffer, California State University, Fresno, Fresno, California, USA

Address correspondence to Evelyn Jantscher-Krenn, evelyn.jantscher-krenn@medunigraz.at, or Christine Moissl-Eichinger, christine.moissl-eichinger@medunigraz.at.

The authors declare no conflict of interest.

See the funding table on p. 26.

The human microbiome is a complex ecosystem of microorganisms, undergoing substantial changes from birth to adulthood (1). Among the various microbiomes,

the oral microbiome is one of the most complex microbiomes and comprises over 700 identified species (2, 3). The oral cavity is a primary entry point for the colonization of both oral and gastrointestinal tract (GIT), making it an accessible site for assessing microbial communities. The unique community of microbes in the oral cavity is in fact very important and any disruption in early oral colonization and the establishment of a healthy oral microbiome is linked with several oral diseases, including dental caries and periodontitis, which could start with the emergence of teeth (4), as well as increased susceptibility to systemic diseases such as cardiovascular disease, due to the presence of potential pro-inflammatory mediators present in periodontium (5).

The formation of the oral microbiome in early childhood is known to be influenced by both host and environmental factors, including genetics, delivery mode, antibiotic use during birth and early infancy feeding mode, and the characteristics of the parental oral microbiome (6). However, the process of initial acquisition and development of this complex microbiome during infancy is not fully understood.

The oral cavity is constantly exposed to oxygen on its surfaces, yet it contains numerous anoxic environments that provide habitats and favorable conditions for anaerobic metabolism and microbial growth. These include biofilms, dental pockets, subgingival crevices, and crypts of the tonsils (7). In general, facultatively anaerobic *Streptococcus* is the predominant early colonizer of the infants' oral cavity, favored by its ability to adhere to epithelial cells (8). By secreting extracellular polymers, it then paves the way for other microbes to emerge, such as *Actinomyces* species (9). The infants' oral microbiome is less diverse compared to adults but becomes more complex within the first month, with mainly *Streptococcus*, *Haemophilus*, *Neisseria*, and *Veillonella* colonizing (4, 8). Nevertheless, knowledge about colonization of non-bacterial microbial members in the oral cavity is very scarce (4).

The oral cavity and gut are connected by the continuous flow of ingested food and saliva through the GIT. Despite this connection, they host distinct microbial communities within their unique microenvironments. Research has shown that these sites harbor locally adapted strains specific to their environments (7, 10, 11), and this segregation is thought to be through various environments including gastric barrier and antimicrobial bile acids within the duodenum. However, little is known about the possible interaction and parallel development of the GIT and oral microbiomes (6, 12, 13). This is particularly true for the radical shifts in the GIT due to the oxygen depletion and the unknown interaction of both environments during this time.

The human GIT in fact harbors the most versatile microbial community. In the initial aerobic phase immediately after birth, the GIT is populated by obligately aerobic or facultatively anaerobic microbes which thrive in the presence of oxygen and are well-adapted to the aerobic environment of the newborn GIT (14–16). The shift to an anaerobic state is driven by oxygen depletion, caused by oxygen consumption by bacteria, colonocytes, or non-biological chemical processes in the cecal contents (17). This step is an essential step in GIT maturation. As oxygen levels decrease, strictly anaerobic bacteria thrive, especially as *Bifidobacterium* species begin to dominate the GIT microbiome. These microbes are adapted to a milk-based diet, using the "bifid shunt" pathway allowing for a fast growth at high lactose concentrations (18). Later, during weaning and introduction of complementary food, other microorganisms replace *Bifidobacterium* species as the dominant microbial group. Steward et al. (19) defined three distinct phases of microbiome progression: a developmental phase at months 3–14, a transitional phase at months 15–30, and a stable phase at months 31–46. These changes are influenced by numerous factors, including birth mode, gestational age, host genetics, environmental factors, and most importantly, feeding mode. The final maturation and stabilization of the GIT microbiome includes not only the settling of highly-oxygen sensitive bacteria, but also methanogenic archaea, which could be indicators for a mature microbiome situation (20).

Similar to fungi, archaea receive less attention regarding their role in the development of a healthy microbiome, although they are present in both the GIT and oral cavity,

often in substantial numbers (21, 22). Few studies have recently shown the detection of archaeal signatures in young infants (22, 23).

Herein, we conducted a longitudinal study on a birth cohort (TRAMIC, https://clinicaltrials.gov/study/NCT04140747) of 30 Austrian infants to investigate the dynamics of aerobic and anaerobic bacteria and archaea in the oral cavity and GIT. The cohort included 15 vaginally delivered infants and 15 born via C-section. Daily up to monthly monitoring of the infants' oral and GIT microbiomes was performed using shotgun metagenomic and amplicon sequencing. This allowed us to assess the development of aerobic and anaerobic microbiomes in parallel at both sites, correlating these patterns with birth mode and infant nutrition.

By elucidating the colonization patterns and ecological dynamics of obligate anaerobes and archaea in both oral and gut environments, this study aims to provide insights into a more fine-tuned early development of the infant microbiome. Understanding the factors shaping microbial colonization during infancy is fundamental for deciphering the role of the microbiome in the course of life and may lead to new strategies to promote infant health and well-being.

## MATERIALS AND METHODS

### Study design

A total of 32 mother-infant pairs were enrolled in the study during their prepartum visits to the Department of Gynecology at the state hospital Graz, Austria before delivery. These participants provided informed consent and obtained oral swabs and stool samples from their infants at various time intervals, commencing immediately after delivery. The primary objective of this pilot study was to investigate the anaerobic microbiome, with a specific focus on archaea, in the oral cavity and GIT of infants throughout their initial year of life.

Detailed inclusion and exclusion criteria can be found in our prior publication (24). In short, every pregnant woman included in the study was in good overall health had no tabacco or alcohol abuse, had not undergone antibiotic treatment within the past 6 months, and was 18 years of age or older. Additionally, their infants were required to be healthy, full-term singletons without any anomalies.

Metadata from all women and infants are listed in the GitHub Repository https://github.com/CharlotteJNeumann/InfantDevelopmentTRAMIC.

In sum, two women opted to discontinue their participation during the study, resulting in 30 infants successfully completing the sample collection phase. Among them, 15 infants were delivered via C-section, while the remaining 15 were born vaginally.

### Sample collection and processing

Oral swabs and stool samples were gathered from all 30 infants at various time points. Stool samples were obtained by spooning the stool from the diaper, avoiding contact with the diaper whereas oral samples were collected by striking the buccal mucosa. Stool samples were collected three times during the initial days of life (S1 [first stool, day 1], S2 [days 2–3], and S3 [days 3–5]). Oral samples were obtained twice during the first days of life (O1 [day 1] and O2 [days 3–5]). Both sample types were collected monthly until the infants reached their first birthday [months 1 (M01) to 12 (M12)]. The collection was performed either by the study nurse at the hospital or by the mothers themselves, following clear instructions on proper collection and storage procedures.

Stool samples were obtained using sterile collection tubes, while oral samples were collected from the buccal mucosa of the cheek using FLOQSwabs (Copan, Milan, Italy). Subsequently, all samples were refrigerated, transported to the laboratory on ice and stored at −80°C until further processing.

Genomic DNA was extracted from the oral swabs utilizing the QIAamp DNA Mini Kit (QIAGEN) with slight modifications: 500 µL of Lysis Buffer (sterile filtered, 20 mM Tris-HCl

at pH 8, 2 mM Na-EDTA, and 1.2% Triton X-100) was added. To all samples, 50 µL of Lysozyme (10 mg/mL, Carl Roth) and 6 µL of Mutanolysin (25 KU/mL, Merck) were added, followed by an incubation at 37°C for 1 h. The resulting mixture was transferred to Lysing Matrix E tubes (MP Biomedicals) for mechanical lysis at 5,500 rpm for 30 s two times using the MagNA Lyser Instrument (Roche, Mannheim, Germany). Following mechanical lysis, the samples were centrifuged at 10,000 × $g$ for 2 min to separate the beads from the supernatant. Subsequently, DNA extraction was performed according to the provided instructions, with the elution of DNA in 60 µL of Elution Buffer.

Stool samples were processed utilizing the QIAamp DNA Stool Mini Kit (QIAGEN) with slight modifications: approximately 200 mg of stool was combined with 500 µL Lysis Buffer (sterile filtered, 20 mM Tris-HCl pH 8, 2 mM Na-EDTA, and 1.2% Triton X-100) and homogenized. To the homogenized samples, 50 µL of Lysozyme (10 mg/mL, Carl Roth) and 6 µL of Mutanolysin (25 KU/mL, Merck) were added and incubated at 37°C for 1 h. Following the incubation, 500 µL Inhibitex was introduced to the samples, homogenized and transferred to Lysing Matrix E tubes (MP Biomedicals) for mechanical lysis at 6,500 rpm for 30 s two times using the MagNA Lyser Instrument (Roche, Mannheim, Germany). After mechanical lysis, the samples were incubated at 70°C for 5 min and then centrifuged for 10,000 × $g$ for 3 min to segregate the beads from the supernatant. The resulting supernatant was then transferred to 2 mL Eppendorf tubes and the remaining steps of the DNA extraction were conducted following the kit protocol. The elution of DNA was carried out using 200 µL of Elution Buffer.

Throughout the DNA extraction procedure, negative controls and mock communities as positive controls were incorporated and processed concurrently.

## PCR amplification

The genomic DNA was used to amplify the V4 region of the 16S rRNA gene employing Illumina-tagged primers, namely 515FB and 806RB (Table 1). To determine the archaeal communities, a nested PCR was performed using the primer combination 344F-1041R/519F-Illu806R, as described previously (25). PCR reactions were performed in triplicate in a final volume of 25 µL, containing TAKARA Ex Taq buffer with MgCl2 (10×; Takara Bio Inc., Tokyo, Japan), primers at 200 nM, dNTP mix at 200 µM, TAKARA Ex Taq Polymerase at 0.5 U, water (Lichrosolv; Merck, Darmstadt, Germany) and DNA template (1–2 µL of genomic DNA) and pooled after amplification. The specific conditions for PCR amplification are listed in Table 2.

## Amplicon sequencing, bioinformatics, and statistical analysis

The library preparation and sequencing of amplicons were conducted at the Core Facility Molecular Biology, Center for Medical Research, Medical University of Graz, Graz, Austria. Briefly, DNA concentrations were normalized using a SequalPrep normalization plate (Invitrogen) and each sample was uniquely indexed through an eight-cycle index PCR with a unique barcode sequence. Following the pooling of these indexed samples, a gel cut was performed to purify the products from the index PCR. Sequencing was executed using the Illumina MiSeq device along with the MS-102-3003 MiSeq Reagent Kit v3-600 cycles (2 × 150 cycles). The generated 16S rRNA gene amplicon data are accessible in the European Nucleotide Archive under the study accession number PRJEB77729.

**TABLE 1** Primer pairs used for universal and archaeal PCRs

| Approach and target | Name | Sequence (5′–3′) | Reference |
|---|---|---|---|
| PCR Universal | 515FB | GTGYCAGCMGCCGCGGTAA | 26 |
| | 806RB | GGACTACNVGGGTWTCTAAT | 26 |
| PCR Archaea I/II | 344F | ACGGGGYGCAGCAGGCGCGA | 27 |
| | 1041R | GGCCATGCACCWCCTCTC | 27 |
| PCR Archaea II/II | 519F | CAGCMGCCGCGGTAA | 27 |
| | 806R | GGACTACVSGGGTATCTAAT | 27 |

**TABLE 2**  PCR settings for the primer pairs used, as already described in reference 24

| Target gene | Primer pair | Initial denaturation | Denaturation | Annealing | Elongation | Final elongation | No. of cycles |
|---|---|---|---|---|---|---|---|
| Universal (16S rRNA gene) | 515FB-806RB | 3 min, 94°C | 45 s, 94°C | 1 min, 50°C | 1 min 30 s, 72°C | 10 min, 72°C | 40 |
| Archaea (16S rRNA gene) | 344F-1041R | 5 min, 95°C | 30 s, 94°C | 45 s, 56°C | 1 min, 72°C | 10 min, 72°C | 25 |
| | 519F-806R | 5 min, 95°C | 40 s, 95°C | 2 min, 63°C | 1 min, 72°C | 10 min, 72°C | 30 |

The analysis of the 16S rRNA gene amplicon data were performed using QIIME2 (28) 2021.1-12 following the previously outlined methodology (29). Quality filtering was performed with the DADA2 algorithm (30) which involved merging paired-end reads, truncation (-p-trunc-len-f 200 -p-trunc-len-r 150) and denoising for the generation of amplicon sequence variants (ASVs). Taxonomic classification (31) was based on the SILVA 138 database (32) and the resultant feature table and taxonomy file were used for subsequent analysis. Contaminating ASVs were identified and eliminated via decontam v1.13 (33) in R (34), running *iscontaminant* in prevalence mode with varying thresholds (oral-bacteria: 0.3; stool-bacteria: 0.3; oral-archaea: 0.5; and stool-archaea: 0.1). Following this, positive controls (mock-communities) and negative controls were excluded from the data sets. Additionally, ASVs classified as chloroplast or mitochondria were removed as well as ASVs with ≤1 read.

For normalization, different approaches were applied for the bacterial and archaeal data sets, taking into account their respective composition. SRS (scaling with ranked subsampling) normalization was run in QIIME2 (28) applying different $c_{min}$ for the bacterial data set (oral-bacteria: $c_{min}$ = 8,400; stool-bacteria: $c_{min}$ = 3,800). The archaeal data sets underwent TSS normalization (total sum normalization). The number of samples subjected to analysis and kept after normalization are listed in Fig. S1.

Several plot types, including stacked bar plots and PCA plots, were generated using MicrobiomeExplorer (35) in R (34).

Differentially abundant taxa were defined by q2-aldex2 (36–38) in QIIME2 (28). To display those taxa in boxplots (packages: ggplot2 (39), dplyr (40), reshape (34, 41), the data of relative abundance were first CLR transformed in R (34).

Alpha-diversity numbers as well as beta-diversity (PERMANOVA) were calculated with the microbiome package (42) in R (34) and plotted with ggplot2 (39) and dplyr (40).

Longitudinal linear mixed-effect models were created with q2-longitudinal (43) in QIIME2 (28) with the option "linear-mixed-effects" for Shannon diversity and "first-distances" additionally for beta-diversity.

## Identification of oxygen requirements

The data sets of universal amplicon data were further investigated regarding the underlying type of respiration. This information had to be collected and entered manually. As resolution from amplicon sequencing is scarce on species level, the genus level was taken into account and physiology data were extracted from bacdive (https://bacdive.dsmz.de/). Therefore, type strain representatives were used, and the common denominator was chosen. We are aware of the problem that physiological data might differ between several species of the same genus; therefore, we handle those data with great care and only as an approximation. In the category of respiration, we assigned three groups: obligate aerobe (listed as "obligate aerobes" and "aerobes"), facultative anaerobe (listed as "microaerophile," "facultative aerobe," and "facultative anaerobe") and obligate anaerobe (listed as "anaerobes" and "obligate anaerobes").

## Source tracking

Source tracking was performed to depict the potential of single ASVs of the oral microbiome (source) to be transferred to the GIT microbiome (sink). Therefore, oral and stool data sets were first merged and then TSS normalized, once for the bacterial approach and once for the archaeal approach. Source Tracking was performed with

SourceTracker2 (44) in QIIME2 (28). Rarefaction of source data (oral) and sink data (stool) and vice versa was performed as advised by SourceTracker2 (44) individually per time point. The rarefaction values are listed in a respective table on GitHub (URL: https://github.com/CharlotteJNeumann/InfantDevelopmentTRAMIC). Additionally, using the "--per_sink_feature_assignments" option in SourceTracker2 (44) on TSS-normalized data sets, we could calculate the origin source of a single taxon. The counts were log-transformed for visualization.

## Network calculations and visualization

To infer genus-level associations, we employed SparCC (45) within the SCNIC tool v.0.5 (Sparse Co-occurrence Network Investigation for Compositional data) (46). SparCC was run on default settings with 1,000 permutations and the multiple testing correction method set to "fdr bh." Co-occurrence events were visualized in Cytoscape v.3.10.1 (47), where nodes represent taxa and edges represent co-occurrences according to the SparCC R values. Stress centrality and other network properties were calculated using Cytoscape. Files of stress centrality for single genera are provided on the GitHub repository (URL: https://github.com/CharlotteJNeumann/InfantDevelopmentTRAMIC).

## Metagenomic data

### Shotgun metagenomic sequencing

We performed shotgun metagenomic sequencing of a subset of infants for a few points (O2, S2, S3, M01, M06, and M12). Sequencing libraries were generated with the TruSeq Nano DNA Library construction kit (Illumina, Eindhoven, the Netherlands) and sequenced on an Illumina NovaSeq 6000 platform (Illumina, Eindhoven, the Netherlands; Macrogen, Seoul, South Korea).

### Metagenomic data processing

The raw reads were processed using the ATLAS v.2.18.0 workflow (48). There, quality control (PCR duplicate removal, quality trimming, host removal, and common contaminant removal) was performed leading to QC reads which were then assembled into high-quality scaffolds using megahit. All parameters used for ATLAS are detailed in the config.yaml file, which is available in the GitHub repository (URL: https://github.com/CharlotteJNeumann/InfantDevelopmentTRAMIC). Genome binning was achieved with maxbin2 v.2.2.7 (49), followed by quality assessment of genome bins with checkM v.1.0.1 (50), bin refinement with DASTool v.1.1.6 (51), dereplication with dRep v.3.5.0 (26), and taxonomic classification of representative metagenomic assembled genomes (MAGs) with GTDB v 2.3.2 (27, 52, 53). Cutoffs for high-quality MAGs were set as follows: completeness >90% and contamination <5%.

Metagenomic data could only be obtained for nine oral samples in total due to the challenging nature of buccal mucosa samples, such as high presence of host DNA contamination. Therefore, no further analyses on metagenomic oral data were possible. Strain tracking was performed in inStrain v.1.5.7 (54) in ATLAS (48) with the following cutoffs: percent_genome_compare: ≥50% and popANI: ≥99.999% as indicated in the documentation of inStrain (55). Functional annotations were also run within the ATLAS pipeline (48). First, Prodigal v.2.6.3 (56) was applied for gene prediction and linclust (57) to cluster redundant genes (minid = 0.9 and coverage = 0.9) (57). The quantification of gene abundance per sample was performed using the combine_gene_coverages function via the BBmap suite v.39.01-1 (58). Employing eggnog-mapper (v.2.0.1) (59, 60) on the EggNOG database 5.0, taxonomic and functional annotations were assigned. KEGG annotations were extracted (61–63) and read counts were implemented and analyzed in R, following https://github.com/metagenomeatlas/Tutorial/blob/master/R/Analyze_genecatalog.Rmd. Annotated gene counts were normalized (size

factor normalization) and tested for differential expression between BF and NBF infants using DESeq2 (63).

### Read-centric metagenome analysis

Species' relative abundances were determined using Kraken2/Bracken (64, 65). Initially, Kraken2 v.2.1.2 (64) was employed to profile the quality-filtered reads from ATLAS v.2.18.0 (48) against the Unified Human Gastrointestinal Genome (UHGG v.2.0.1) (66) database of bacterial and archaeal genomes. Subsequently, Bracken v.2.7 (65) was used with default settings to analyze the Kraken2 output and calculate the relative abundance of bacterial and archaeal species. The resulting report files were merged to generate an abundance table of microbial species for further analysis.

## Additional tools used in the manuscript

ChatGPT.com and deepl.com were used for language checks, but not for interpreting the data.

An overview of the available data is displayed in two figures: Fig. S1 is following the STORM guideline and was created with drawio.com (URL: https://drawio.com). Figure S2 displays the data available per sample and individual.

## Reproducibility

We conducted a prospective pilot study, whereas the sample size was not predetermined beforehand. Randomization and blinding of the investigators were not foreseen in the chosen study setup. A full study flow chart is provided in Fig. S1 and S2. Participants 13 and 17 were excluded from the study due to incompleteness. Overall, the study is only partially reproducible, as the data are dependent on the study cohort, which was only sampled once within this study, and sampling of cohorts at the same time window cannot be repeated. However, starting from the raw sequencing data, the analysis is fully reproducible, and all required data, scripts, and details are provided. The STORMS Checklist can be found in the GitHub Repository (URL: https://github.com/CharlotteJNeumann/InfantDevelopmentTRAMIC).

## RESULTS

### Overview on the study population and sample description

Infancy is a dynamic period for microbiome development, with the first 1,000 days of life being the most critical period (67). We highlight the dynamics of microbiome composition and co-occurrence patterns in oral and GIT microbiomes in the first year of life and their transmission patterns, with a focus on the dynamics of anaerobic microorganisms.

Oral and stool samples of 30 infants born either spontaneous ($n = 15$) or via C-section (CS) ($n = 15$) were collected at different time points (Fig. S2). Stool samples were initially collected at three time points (tps) (S1 [first stool, meconium, day 1], S2 [meconium, days 2–3], and S3 [days 3–5]), while oral samples were obtained at two-time intervals (O1 [day 1, prior to feeding, immediately after delivery] and O2 [days 3–5]). Both sample types were collected monthly until 1 year of age (months M01 to M12). The characteristics (covariates) of the two study groups (spontaneous and CS) did not significantly differ (Chi-square test, $P > 0.5$) except for gestational age, which is significantly lower in infants born via CS (Chi-square test, $P < 0.001$). No covariates for study groups of infants that were breastfed (BF) less or longer than 6 months differed significantly except for formula feeding. The metadata of the studied cohort can be found in Tables 3 and 4; Fig. 1. For maximal resolution of the impact of the feeding regimen, the breastfeeding status was assessed for each single time point (month) individually for each infant, leading to dynamic groupings that changed over time. For example, infants who were BF at month 3 (M03) were placed in the BF group for that time point, but once they were no longer BF, they were moved to the "non-breastfed (NBF)" group from that point onward.

**TABLE 3** Perinatal and postnatal factors between spontaneous and C-section delivery

| Individuals | Characteristics | Total (n = 30) | Mode of delivery | | |
|---|---|---|---|---|---|
| | | | Spontaneous (n = 15) | C-section (n = 15) | P value |
| Infants | Sex | | | | 0.46 |
| | Male | 16 (53.3) | 10 (62.5) | 6 (42.9) | |
| | Female | 14 (46.7) | 6 (37.5) | 8 (57.1) | |
| | Antibiotic usage during the first 12 months | | | | 0.66[b] |
| | Yes | 6 (20.0) | 4 (25.0) | 2 (14.3) | |
| | No | 24 (80.0) | 12 (75.0) | 12 (85.7) | |
| | Solid food introduction | | | | 0.72[b] |
| | At 6 months | 12 (40.0) | 7 (43.8) | 5 (35.7) | |
| | Later than 6 months | 18 (60.0) | 9 (56.3) | 9 (64.3) | |
| | Breastfeeding during the first 12 months | | | | |
| | Ever breastfed | 26 (86.7) | 15 (93.8) | 11 (78.6) | 0.32[b] |
| | breastfed after 6 months | 22 (73.3) | 13 (81.3) | 9 (64.3) | 0.42[b] |
| | Formula milk feeding during the first 12 months | | | | |
| | Ever formula fed | 17 (56.7) | 9 (56.3) | 8 (57.1) | 1.00 |
| | Formula fed before 6 months | 9 (30.0) | 3 (18.8) | 6 (42.9) | 0.24[b] |
| | Pet at home | | | | |
| | All types | 11 (36.7) | 5 (31.3) | 6 (42.9) | 0.71 |
| | Fur pet | 10 (33.3) | 5 (31.3) | 5 (35.7) | 1.00[b] |
| | Gestational age (weeks) [mean ± SD] | 39.3 ± 1.2 | 40.1 ± 0.5 | 38.4 ± 1.0 | <0.001[a] |
| | Birthweight (kg) [mean ± SD] | 3.4 ± 0.4 | 3.5 ± 0.3 | 3.4 ± 0.5 | 0.54 |
| | Length of hospital stay (days) [mean ± SD] | 3.7 ± 1.2 | 3.7 ± 1.5 | 3.8 ± 0.7 | 0.82 |
| Mothers | Age at infant's birth | | | | 0.19[b] |
| | <31 years | 8 (26.7) | 5 (31.3) | 3 (21.4) | |
| | 31–35 years | 10 (33.3) | 7 (43.8) | 3 (21.4) | |
| | >35 years | 12 (40.0) | 4 (25.0) | 8 (57.1) | |
| | Gravida | | | | 0.23[b] |
| | <2 | 8 (26.7) | 6 (37.5) | 2 (14.3) | |
| | >2 | 22 (73.3) | 10 (62.5) | 12 (85.7) | |
| | Parity | | | | 0.058[b] |
| | <2 | 10 (33.3) | 8 (50.0) | 2 (14.3) | |
| | >2 | 20 (66.7) | 8 (50.0) | 12 (85.7) | |
| | Abortion | | | | 0.66[b] |
| | 0 | 24 (80.0) | 12 (75.0) | 12 (85.7) | |
| | 1 | 6 (20.0) | 4 (25.0) | 2 (14.3) | |
| | Pre-pregnancy weight (kg) [mean ± SD] | 62.8 ± 11.9 | 64.1 ± 11.4 | 61.4 ± 12.9 | 0.55 |
| | Pre-pregnancy BMI | | | | 0.23[b] |
| | <18.5 | 3 (10.0) | 1 (6.3) | 2 (14.3) | |
| | 18.5–24.9 | 21 (70.0) | 10 (62.5) | 11 (78.6) | |
| | 25.0–29.9 | 6 (20.0) | 5 (31.3) | 1 (7.1) | |

[a]Significance level at P < 0.05.
[b]Chi-square test with more than 20% with less than five counts.

Breastfeeding is considered the most significant microbiome covariate within the first year of life (19). In line with this, in our data set, we found that the feeding type significantly impacted four and one time points for oral and GIT samples, respectively (PERMANOVA; oral: $P < 0.05$ for four tps [M03, M04, M08, and M09]; stool $P < 0.05$ for one tp [M10]) but birth mode impacted only one timepoint for both sample types [PERMANOVA; oral: $P < 0.05$ for one tp [M03]; stool $P < 0.05$ for one tp [M01]). Based on this observation, we mainly focused on the feeding types and their impact on the anaerobic microbiome in the oral cavity and GIT and their transitional phase.

**TABLE 4** Perinatal and postnatal factors between infants that were breastfed less or longer than 6 months

| Individuals | Characteristics | Total (n = 30) | Feeding | | |
|---|---|---|---|---|---|
| | | | Breastfeeding less than 6 months (n = 8) | Breastfed longer than 6 months (n = 22) | P value |
| Infants | Sex | | | | 1.00[b] |
| | Male | 16 (53.3) | 4 (50.0) | 12 (54.5) | |
| | Female | 14 (46.7) | 4 (50.0) | 10 (45.5) | |
| | Antibiotic usage during the first 12 months | | | | 1.00[b] |
| | Yes | 6 (20.0) | 1 (12.5) | 5 (22.7) | |
| | No | 24 (80.0) | 7 (87.5) | 17 (77.3) | |
| | Solid food introduction | | | | 0.42[b] |
| | At 6 months | 12 (40.0) | 2 (25.0) | 10 (45.5) | |
| | Later than 6 months | 18 (60.0) | 6 (75.0) | 12 (54.5) | |
| | Mode of delivery | | | | 0.68[b] |
| | Spontaneous | 26 (86.7) | 3 (37.5) | 12 (54.5) | |
| | C-section | 22 (73.3) | 5 (62.5) | 10 (45.5) | |
| | Formula milk feeding during the first 12 months | | | | |
| | Ever formula fed | 17 (56.7) | 8 (100) | 9 (40.9) | 0.004[a,b] |
| | Formula fed before 6 months | 9 (30.0) | 8 (100) | 1 (4.5) | <0.001[a,b] |
| | Pet at home | | | | |
| | All types | 11 (36.7) | 4 (50.0) | 7 (31.8) | 0.42[b] |
| | Fur pet | 10 (33.3) | 3 (37.5) | 7 (31.8) | 1.00[b] |
| | Gestational age (weeks) [mean ± SD] | 39.3 ± 1.2 | 38.9 ± 1.6 | 39.5 ± 1.0 | 0.32 |
| | Birthweight (kg) [mean ± SD] | 3.4 ± 0.4 | 3.4 ± 0.5 | 3.4 ± 0.3 | 0.96 |
| | Length of hospital stay (days) [mean ± SD] | 3.7 ± 1.2 | 4.4 ± 0.7 | 3.5 ± 1.2 | 0.10 |
| Mothers | Age at infant's birth | | | | 0.56[b] |
| | <31 years | 8 (26.7) | 1 (12.5) | 7 (31.8) | |
| | 31–35 years | 10 (33.3) | 3 (37.5) | 7 (31.8) | |
| | >35 years | 12 (40.0) | 4 (50.0) | 8 (36.4) | |
| | Gravida | | | | 0.64[b] |
| | <2 | 8 (26.7) | 3 (37.5) | 5 (22.7) | |
| | >2 | 22 (73.3) | 5 (62.5) | 17 (77.3) | |
| | Parity | | | | 1.00[b] |
| | <2 | 10 (33.3) | 3 (37.5) | 7 (31.8) | |
| | >2 | 20 (66.7) | 5 (62.5) | 15 (68.2) | |
| | Abortion | | | | 1.00[b] |
| | 0 | 24 (80.0) | 7 (87.5) | 17 (77.3) | |
| | 1 | 6 (20.0) | 1 (12.5) | 5 (22.7) | |
| | Pre-pregnancy weight (kg) [mean ± SD] | 62.8 ± 11.9 | 65.3 ± 16.7 | 61.9 ± 10.1 | 0.98 |
| | Pre-pregnancy BMI | | | | 0.86[b] |
| | <18.5 | 3 (10.0) | 1 (12.5) | 2 (9.1) | |
| | 18.5–24.9 | 21 (70.0) | 5 (62.5) | 16 (72.7) | |
| | 25.0–29.9 | 6 (20.0) | 2 (25.0) | 4 (18.2) | |

[a]Significance level at P < 0.05.
[b]Chi-square test with more than 20% with less than five counts.

## The oral cavity and GIT are rapidly exposed to strict anaerobes

Samples taken right after birth (labeled "O1" and "S1") and within the first days of life (labeled "S2", "S3," and "O2") showed that newborns get colonized rapidly by various microbes. The first obligate anaerobic bacteria detected in the oral cavity and GIT were *Rothia* (oral), *Streptococcus*, *Staphylococcus* (both oral and GIT), *Bifidobacterium*, and *Enterococcus* (GIT).

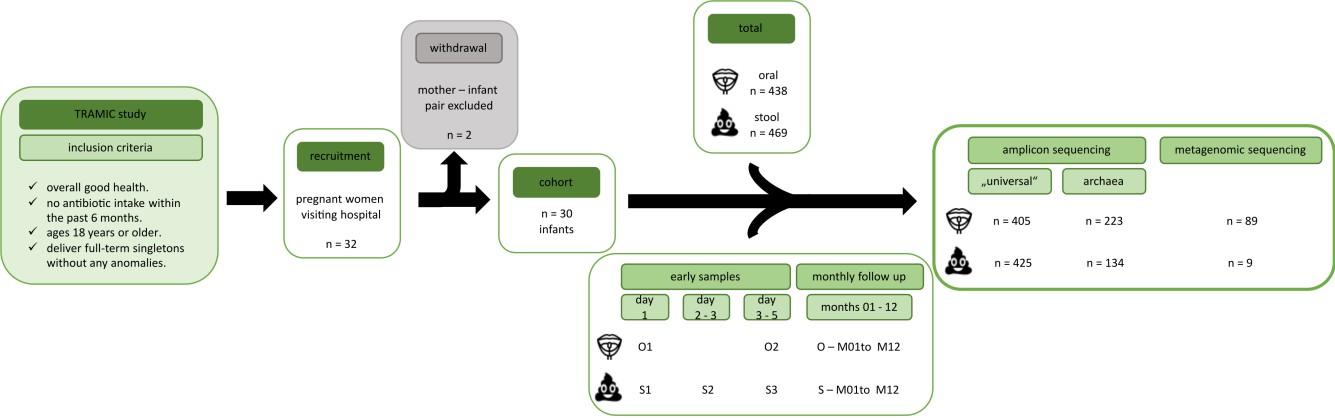

**FIG 1** Graphical abstract of the study including recruitment, sample collection and genomic data received.

Interestingly, next to bacteria, archaeal signatures could also be detected in those early samples (Fig. S3). Archaeal diversity was higher in those early-stage samples with *Methanobrevibacter*, *Methanobacterium*, *Methanosphaera*, and *Methanocorpusculum* (all obligate anaerobes) being present next to unclassified Woesearchaeales (oral and GIT) and unclassified Nitrososphaeraceae (GIT). For the latter two, the oxygen requirements are unknown, as these taxa were not classified deep enough. At M01, *Methanobrevibacter* was predominant amongst archaea. As expected, samples collected at the very early stages showed different microbial profiles compared to the ones collected at M01, revealing a shift from S1/O1 to M01 (Fig. S3). It is assumed that the first samples taken immediately after birth do not reflect the inhabitant microbial community, but rather a microbial contamination given the sterile environment in the womb (68). Although the microbial ecosystem may not be fully functional at this time stage, microbial colonization can already start with exposure to the extra-uterine environment and subsequent oral-GIT transmission. However, the main analyses drawn out in this paper focus on samples collected at M01 and later, when microorganisms have started to establish.

## *Staphylococcus* and *Streptococcus* are early but transient colonizers of the oral cavity

In the first month of life, the human skin (parents and family members) is an important source of microbial influx from the environment (8, 69). This is underlined by our data, showing high relative abundances of *Staphylococcus* (facultatively anaerobic) representing a taxon that is mainly skin- (and mucosa) associated (Fig. 2a). We did not find significant differences in the relative abundance of *Staphylococcus* between BF and NBF infants (Aldex2, all tps, $P > 0.05$, Fig. S4), indicating a general substantial transfer from skin to the oral cavity, independent of feeding mode.

To assess the connectivity and co-occurrence of microbes, we built networks for each time point by forming modules in SCNIC at the genus level (Fig. 2b; Fig. S5). From M03 on (Fig. S5), *Staphylococcus* has a very minor relative abundance and appears only sporadically in the co-occurrence networks with low centrality compared to other players (Fig. 2b and Fig. S5 for the complete networks; stress centrality BF: M02: 4, M04: 8, M06: 4; M07: 32; NBF: M01: 150, M02: 28, M05: 82, M10: 44, and M12: 28), indicating its transient colonization in the oral cavity of infants in early life.

Especially in the first month of life, the infant's early oral microbiome is predominated by facultatively anaerobic *Streptococcus* (Fig. 2a). Interestingly, the centrality of *Streptococcus* in microbial networks is surprisingly low, although the abundance is very high (>60% relative abundance) (Fig. 2b). This indicates that even if a microbe is very abundant, this does not necessarily mean that it is an important player in the networks formed by the microbial community. It appears that streptococci do not interact with

other microbes on a large scale, but rather rely on themselves and act independently. Streptococci, which are mainly involved in carbohydrate metabolism, are considered pioneer species that lead the assembly of a complex oral microbiome (70). The dominance of *Streptococcus* is higher in BF infants, reflected by both relative abundance (Aldex2, M04, M05, M06, and M09: $P < 0.05$, $q > 0.05$, Fig. S4; Fig. 2a) as well as network centrality (Fig. 2b; Fig. S5, tp M04 and M06 as an example: BF: M04 = 48,

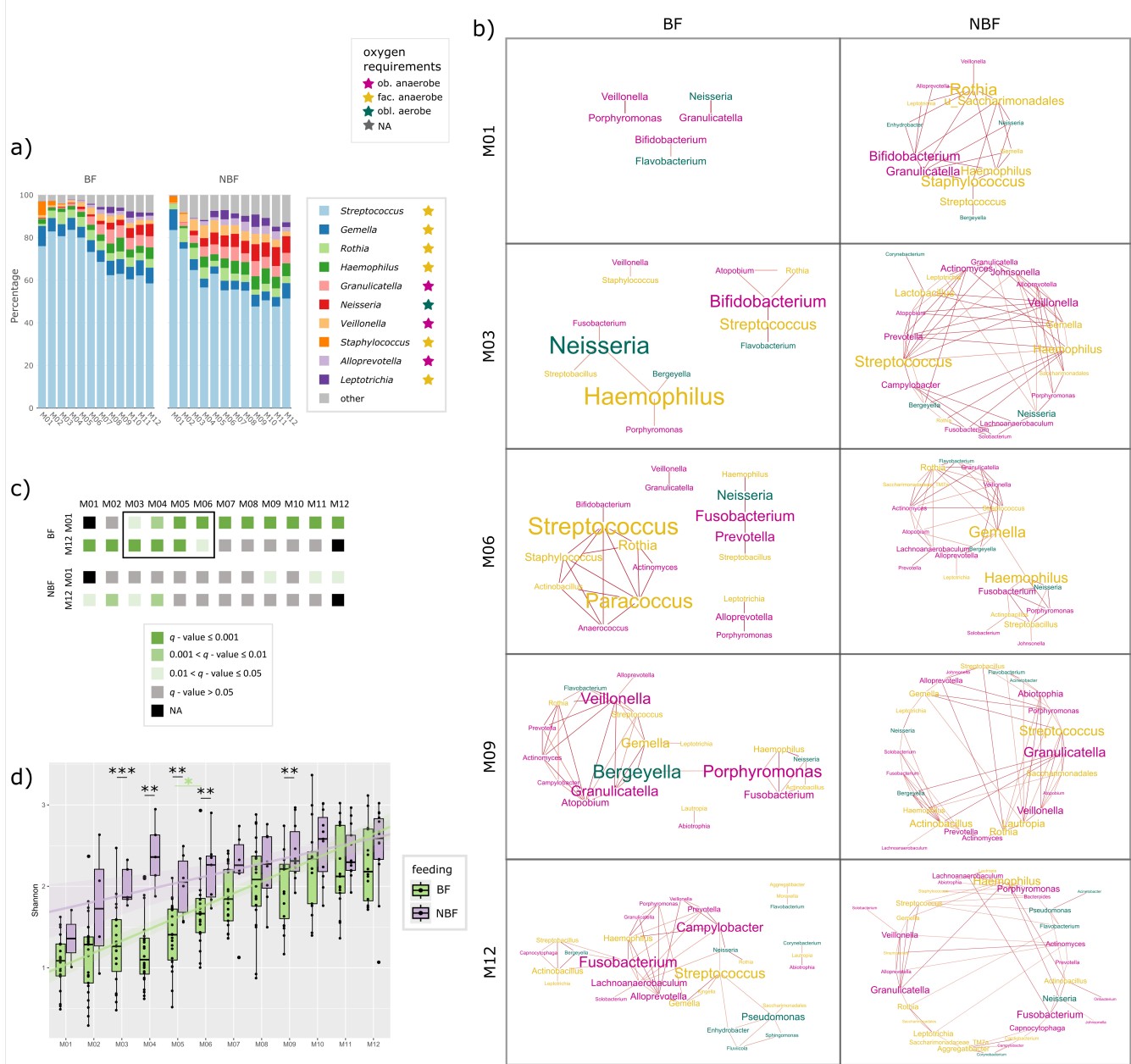

**FIG 2** Panel on the oral microbiome. (a) Stacked bar chart showing the relative abundance of the 10 most common bacterial genera in the oral cavity per time point (months M01 to M12). Data are shown separately for breastfed (BF) and non-breastfed (NBF) infants. The oxygen requirement of the respective genera is highlighted by colored stars: pink: obligate anaerobes, yellow: facultative anaerobes, and petrol: obligate aerobes. (b) Networks on oral samples of BF and NBF infants of selected time points (months M01, M03, M06, M09, and M1). Font size indicates stress centrality, colors indicate oxygen requirement: pink: obligate anaerobes, yellow: facultative anaerobe, and petrol: obligate aerobes. (c) Pairwise beta-diversity comparisons (PERMANOVA) from months M01 to M12 to all other time points to depict the transition phase. Gray and shaded green colors indicate $q$-values; data are separated for BF and NBF infants. (d) Shannon diversity of oral samples depicted for BF (= light green) and NBF (= lavender) infants with asterisks indicating significant differences ($q$-values) between those two groups. Significant $q$-values between tps in BF infants are indicated with light green asterisks.

M06 = 18; NBF M04 = 128, and M06 = 88). *Streptococcus* shows a decrease in relative abundance starting from M05 onwards (Fig. 2a; Aldex2, *P* > 0.05 at any tp pairwise comparison).

## Distinct transitional phases of the oral microbiome in BF infants

Taking several analyses into account (alpha-diversity, beta-diversity, Aldex2, and networks), we were able to outline a time frame in which the oral microbiome changes the most and which thus represents a transition phase of the oral microbiome towards a more mature microbial community.

When the beta-diversity of all oral samples from BF infants was compared with the first (M01) and last time points (M12), it was found that the samples from M03 to M06 differed significantly from these reference points (PERMANOVA; Fig. 2c). Interestingly, this effect was way less pronounced in NBF infants, where only the first 4 months are significantly different to M12 (Fig. 2c), indicating a less defined maturation period in this group.

These observations go hand in hand with patterns we observed in alpha-diversity and allowed for defining the transitional phase of the oral microbiome. This phase seems to be less marked and more gradually processing in NBF than in BF infants. Alpha-diversity was overall significantly increasing within the first year of life (Shannon Fig. 2d and evenness and richness, Fig. S6) (Shannon, longitudinal linear mixed-effect model (LME), *P* < 0.001, Fig. S7) and was higher in NBF infants than in BF infants (LME, *P* = 0.002, Fig. S7). NBF infants' alpha-diversity was more rapidly increasing within the first 4 months of life (M01 to M04) (Fig. 2d). Highest differences in alpha-diversity between BF and NBF infants were observed at M03–M06 and M09 (Shannon diversity, *t* test: M03 *q* = 0.00044, M04 *q* = 0.0013, M05 *q* = 0.0079, M06 *q* = 0.0081, and M09 *q* = 0.0096; evenness, *t* test: M03 *q* = 0.0056, M04 *q* = 0.00052, M05 *q* = 0.029, M06 *q* = 0.0098, M09 *q* = 0.0096; richness, Wilcoxon: M05 *q* = 0.0055). For BF infants, a significant increase in both Shannon diversity and richness could be observed from M05 to M06 (Wilcoxon, *q* = 0.0055, Fig. 2d; Fig. S6).

As already pointed out above, the decrease of *Streptococcus* also started earlier in NBF than in BF infants, supporting the observation of two different dynamics patterns of microbiome maturation in BF and NBF infants.

During this transitional phase, in contrast to *Staphylococcus* and *Streptococcus* which were declining, several genera increased in relative abundance. A distinction can be made between genera such as *Gemella*, *Rothia*, and *Haemophilus* that were already abundant in the first month (all facultative anaerobes) and genera such as *Granulicatella* (obligate anaerobe), *Neisseria*, *Veillonella* (obligate anaerobe), *Alloprevotella* (obligate anaerobe), and *Leptotrichia* which were newly introduced (Fig. 2a). During the transition phase on BF infants (M03–M06), *Streptococcus* did not form any co-occurrence connections with those "new" genera *Neisseria*, *Alloprevotella*, or *Leptotrichia* at all, except with *Neisseria* at M12 (Fig. 2b). After M07, *Alloprevotella* and *Leptotrichia* began to co-occur indirectly with *Streptococcus*, primarily with *Gemella* serving as the connecting node. This suggests that *Gemella* may mediate the integration of co-occurrence between *Alloprevotella*–*Streptococcus* and *Leptotrichia*–*Streptococcus*. This integration could exemplify how the introduction of a new microbe (*Alloprevotella*/*Leptotrichia*) into the community may be facilitated by an existing microbe (*Gemella*), contributing to community maturation. Whereas niche-sharing between *Streptococcus* and the genera that were fairly abundant already at the beginning, *Gemella* and *Rothia,* were common independent of feeding mode, co-occurrence of *Streptococcus* with *Haemophilus* was exclusive to NBF infants. Additionally, *Streptococcus* showed intensive co-occurrence connections with *Granulicatella* and *Veillonella*.

These "new" bacterial genera showed a lagged increase in relative abundance in BF infants between mainly M04 and M05 (Fig. S4; Aldex2, *Granulicatella P* = 0.003, *Neisseria P* = 0.012, *Veillonella P* = 0.003, *Leptotrichia P* = 0.002). This supports our findings of a lagged transitional phase between BF and NBF infants (can also be seen in Fig. 2a).

## Breastmilk maintains simplicity of oral microbial network structures

Starting around M07, after the transitional phase of the BF infants' oral microbiome had concluded, the microbiomes of BF and NBF infants became more similar in terms of alpha-diversity, beta-diversity, and differentially abundant taxa, with fewer significant differences observed on genus level. This coincides with the time when solid food typically constitutes a large portion of the infants' diet, suggesting that solid food acts as a leveling factor for the oral microbiomes of BF and NBF infants.

A notable difference between the oral microbiomes of BF and NBF infants was the overall organization of the microbial networks (Fig. 2b). From the beginning, the microbial networks in NBF infants were very complex, with multiple nodes (microbial genera) and edges (co-occurrences between genera). Several genera exhibited high-stress centrality, meaning they co-occurred with multiple genera and thus were central in the network. This structure changed only slightly over time. In contrast, the oral microbial network of BF infants was much less complex, with fewer microbial players sharing a similar niche. These differences in complexity were especially pronounced in the first month until about M06. This could be attributed to the different nutrient compositions of breast milk (BM) and formula milk, with BM being digested very efficiently, leading to a simplified microbial community.

After M06, the networks in BF infants became more complex, with higher stress centrality of individual microbes (from <50 until M06 to 800 at M11, Fig. S8) and more microbial members co-occurring (18 nodes at M06 to 31 nodes at M12). This likely reflects the introduction of solid food, which provokes a new mode of microbial (inter-)action. However, despite solid food becoming a major component of the diet of a 1-year-old child, the administration of BM, even in smaller proportions, still seemed to influence the GIT microbiome. Even at M12, clear differences between the microbial networks of BF and NBF infants were still evident (Fig. 2b).

## *Neisseria* and its key role for the thriving of obligate anaerobes in the oral cavity

Within the first year of life, the relative abundance of facultative anaerobes decreased from 96% to 76%, with a corresponding increase in obligate anaerobes and obligate aerobes (Fig. S9). NBF infants exhibited a higher load in obligate anaerobes in their oral cavity compared to BF infants (Fig. S9). Despite the high oxygen exposure in the oral cavity, various microenvironments and mechanisms, such as biofilm formation, create suitable niches for obligate anaerobic microbes.

A key player in the infant's oral microbiome is *Neisseria*, an obligate aerobe that plays a major role in biofilm-based oral microbiome networks. In fact, *Neisseria* can protect obligate anaerobes from oxygen (71), likely by consuming it through respiration.

In our networks, *Neisseria* was present at all tps, primarily co-occurring with obligate anaerobes (e.g., *Porphyromonas*, *Fusobacterium*, and *Lachnoanaerobaculum*) and facultative anaerobes (e.g., *Haemophilus*, *Streptobacillus*, and *Leptotrichia*), but not with other obligate aerobes such as *Bergeyella* (Fig. 2b). Typically, interactions between *Streptococcus* species and *Veillonella* were found during the early stages of oral biofilm formation (72). Interestingly, in the networks of NBF infants, we also observed directed nodes between *Neisseria* and obligate aerobic genera such as *Flavobacterium* (M02) and *Bergeyella* (M07). *Neisseria* was more abundant in NBF infants (Aldex2 at M04–M07, M09, and M11 Fig. S4) and showed high centrality in the networks (Fig. 2b), indicating its prominent role in the oral microbiome of NBF infants.

## Archaeal signatures are rare and probably transient in the oral microbiome of the first year of life

A high-resolution (taxonomy), highly specific nested PCR approach was used to detect the taxonomic diversity of archaea. The method was successful for 224 out of 415 oral samples which gave a high-quality amplicon result (see also overview Fig. S2).

*Methanobrevibacter* was indicated to be the dominant archaeal player in the oral niche (Fig. 3a). All infants harbored archaeal signatures in their oral cavity in at least one tp (Fig. 3b). The sporadic loss and emerge of these archaeal signatures in the oral cavity underline our hypothesis that archaea are transient and dependent on environmental input, and we could not define any longitudinal development pattern.

Besides *Methanobrevibacter*, unclassified Woesearchaeales could be detected in several infants and time points, followed by *Methanobacterium*, unclassified Nitrososphaeraceae [probably stemming from human skin (21)] and *Candidatus* Nitrosotenuis.

Given that a nested PCR approach is unfavorable for drawing conclusions about abundance, statistics for the archaeal genera were only performed for their presence/absence using Fisher's *t* test. No significant differences were found for feeding type or birth mode at any time point. As such, we conclude that young infants do not carry a stable oral archaeome.

## The influence of the oral microbiome on the GIT microbiome decreases within the first year

To evaluate the potential of the oral microbiome as a source of microbes transferred to the GIT over the first year of life, we conducted source tracking analyses. Overall, the oral microbiome contributed minimally to the GIT microbiome with the highest contribution at tp M01 (mean 18, 27% probability) gradually decreasing over time (mean 7.63% probability at M12) (Fig. 4a).

We also calculated the origin source of individual taxa. An overview of the top 30 ASVs (Fig. 4b) showed that at M01, various ASVs found in the GIT were derived from the oral cavity (see below). This could be attributed to the GIT's limited and unstable colonization by microbes at this early stage, making it more susceptible to influence of the oral microbiome. Additionally, the gastric barrier may not be fully developed at this stage, allowing more microbial transmission from the oral cavity to the GIT.

The main representatives of this early transmission were *Bifidobacterium*, *Staphylococcus*, and *Streptococcus*. *Streptococcus* is the primary genus transferred from oral cavity to the GIT, with one dominant ASV being constantly transmitted over the first year (Fig. 4). In contrast, one *Haemophilus* ASV (ASV06) gained source tracking potential starting from M07. *Bifidobacterium* showed a notable peak at tp M07 and M08 (Fig. 4b).

It is notable that the genera of ASVs tracked from the oral cavity to the GIT generally play central roles in microbial networks or exhibit high abundance. Running source tracking in reverse mode (from GIT [source] to oral [sink]), indicated a number of ASVs shared between oral cavity and GIT: *Bifidobacterium* ASV01, *Haemophilus* ASV06 and ASV16, *Lactobacillus* ASV26, *Rothia* ASV15 and ASV24, *Staphylococcus* ASV07 and ASV27, *Streptococcus* ASV02–ASV05, and *Veillonella* ASV18 (Fig. S10 and S11).

Source tracking was further performed for the archaeal data set (nested PCR approach, based on presence/absence). It was found that the oral microbiome cannot be considered a potent source for the GIT archaeome, as only in three samples a minimal contribution 0.1% (i29_M02), 0.3% (i23_M06), and 0.3% (i05_M12) was detected.

## The GIT microbiome develops more steadily throughout the first year of life than the oral microbiome

Similar to the oral microbiome, stool samples from BF infants exhibited a distinct but prolonged transition period from M03 to M08 (PERMANOVA, Fig. 5a). Again, no such obvious time frame was observed for NBF infants, highlighting fewer differences in the GIT microbiome composition between the start and end points of comparison. This is further illustrated by generally smaller Bray-Curtis distances in NBF infants compared to BF infants (Fig. S12).

This pattern is also reflected by our microbial networks and alpha-diversity. Similar to the oral microbiome, Shannon diversity, evenness, and richness of the GIT microbiome increased over time within the first year of life (LME for Shannon, $P = 0.004$, Fig. S13), with a

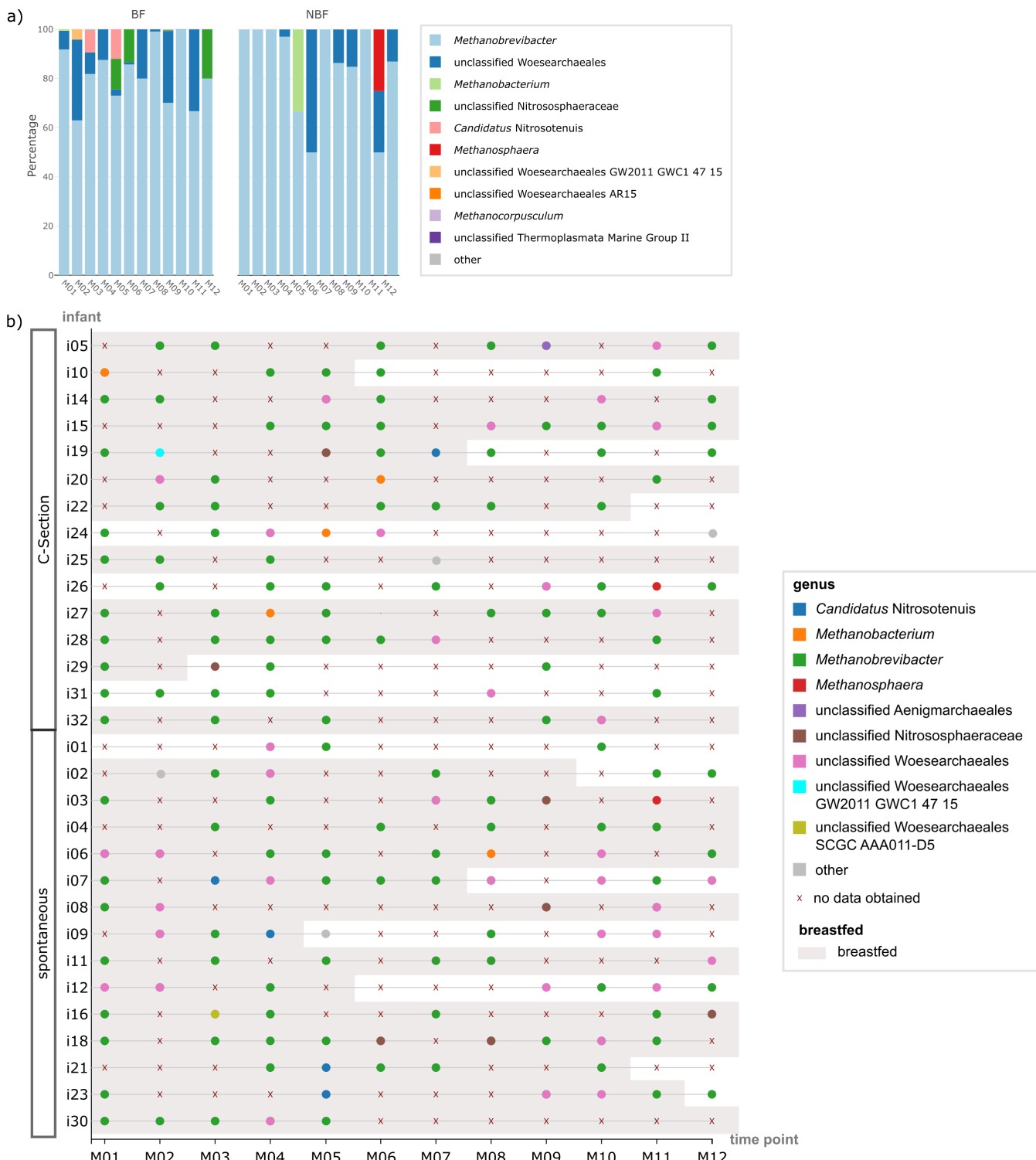

**FIG 3** Panel on the oral archaeome. (a) Stacked bar chart showing the relative abundance of the 10 most common archaeal genera in the oral cavity per time point (months M01 to M12); data are shown separately for breastfed (BF) and non-breastfed (NBF) infants. (b) Beeswarm plot on absence/presence of archaeal signatures in the oral microbiome per infant and time point (months M01 to M12); time points at which the infants were BF are underlaid with gray and infants are sorted by their mode of delivery.

more pronounced increase in BF infants compared to NBF infants (Shannon diversity, LME *P* < 0.001). However, these changes were not significantly different between individual tps

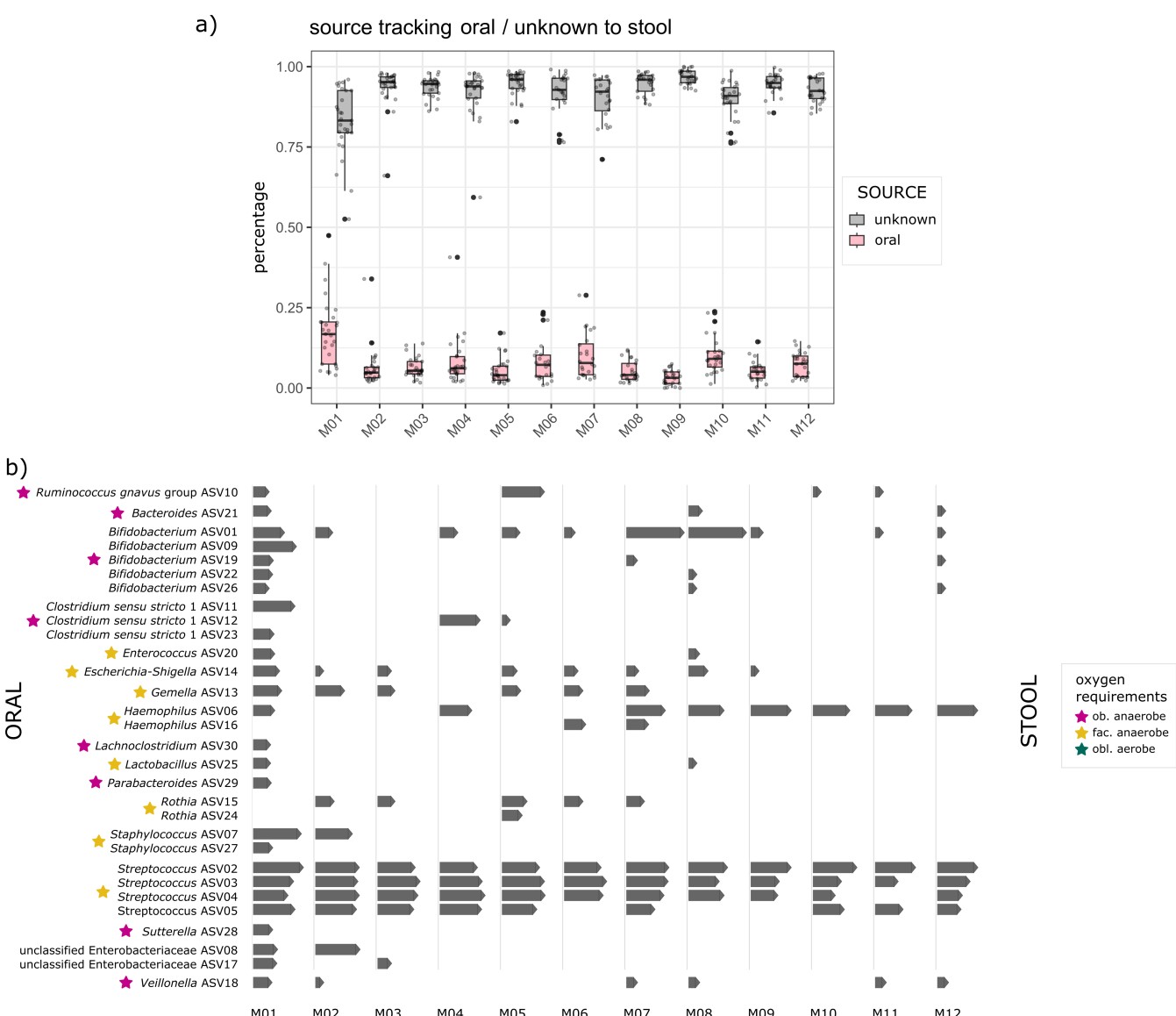

**FIG 4** Source tracking from oral to GIT. (a) Source tracking probability of bacterial taxa being transferred from oral and unknown sources to the GIT as a sink, depicted per time point (months M01 to M12). (b) Source tracking of specific bacterial taxa being transferred from oral source to GIT sink. The top 30 ASVs are depicted per time point (months M01 to M12), the length of the bars indicates the log-transformed counts of a taxon. The oxygen requirement of the respective ASV is highlighted by colored stars: pink: obligate anaerobes, yellow: facultative anaerobes, and petrol: obligate aerobes.

(Fig. 5b; Fig. S14). In general, alpha-diversity and mainly Shannon diversity was again slightly higher in NBF infants than in BF infants (Fig. S13, *t* test: M04: *q* = 0.029 M05: *q* = 0.044; richness, Wilcoxon: M03 *q* = 0.019, M05: *q* = 0.021, Fig. S14).

## The GIT microbiome stabilizes by establishing complex anaerobic microbial communities

Within the first year of life, a very complex microbial network was established (BF M012: nodes *n* = 45, edges *n* = 78, average number of neighbors *n* = 3.467; NBF M12: nodes *n* = 38, edges *n* = 56, and average number of neighbors *n* = 3.056). In the first month, especially for BF infants, only few bacterial genera were found to co-occur, and stress centrality in general was low (Fig. S15). As alpha-diversity increased over time, the number of genera included in networks also grew. The microbial networks in the NBF

infants were more complex from the early months with a higher number of nodes and edges, indicating more microbial interactions compared to BF infants (Fig. 5c).

The GIT microbiome was predominated by various obligately anaerobic microbes at all tps and their relative abundances were constantly increasing within the first year of life (Fig. S16, from ~50% in M01 to ~70% in M12). Initially, the human GIT contains little amounts of oxygen which is gradually consumed by microbial activity. Indeed,

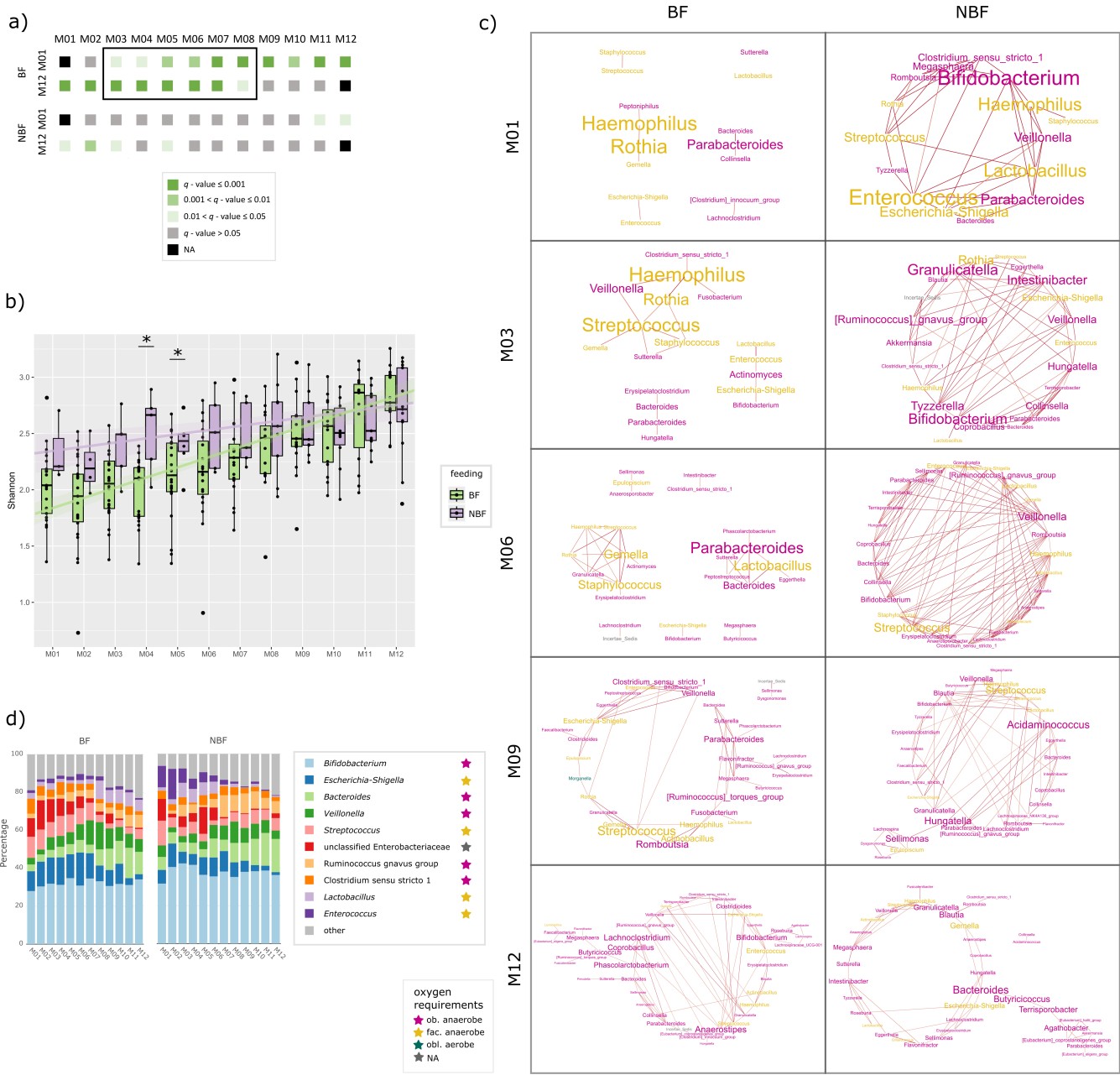

**FIG 5** Panel on the GIT microbiome. (a) Pairwise beta-diversity comparisons (PERMANOVA) from months M01 to M12 to all other time points to depict the transition phase. Gray and shaded green colors indicate $q$-values; data are separated for breastfed (BF) and non-breastfed (NBF) infants. (b) Shannon diversity of stool samples depicted for BF (= light green) and NBF (= lavender) infants with asterisks indicating significant differences ($q$-values) between those two groups. No significant $q$-values between tps within one group (BF or NBF). (c) Networks on stool samples of BF and NBF infants of selected time points (months M01, M03, M06, M09, and M12) Font size indicates stress centrality, colors indicate oxygen requirement: pink: obligate anaerobes, yellow: facultative anaerobe, and petrol: obligate aerobes. (d) Stacked bar chart showing the relative abundance of the ten most common bacterial genera in the GIT per time point (months M01 to M12). Data are shown separately for BF and NBF infants. The oxygen requirement of the respective genera is highlighted by colored stars: pink: obligate anaerobes, yellow: facultative anaerobes, and petrol: obligate aerobes.

in the first month of life, some facultatively anaerobic bacteria were still detectable with central roles in bacterial networks (Fig. 5c and all networks in Fig. S16 and S17), including taxa of the genera *Escherichia-Shigella*, *Rothia*, *Haemophilus*, *Staphylococcus*, *Enterococcus*, *Lactobacillus*, and *Gemella*. This was particularly evident in BF infants, with *Haemophilus* showing particularly high-stress centrality. In contrast, we hardly found any obligate aerobes in the GIT, correlating with very low oxygen levels after initial oxygen consumption (Fig. S16).

The most prominent obligate anaerobe in the GIT was *Bifidobacterium* (Fig. 5d), consistently representing about 30% (relative abundance) at all tps. Similar to *Streptococcus* in the oral microbiome, *Bifidobacterium* was, beyond its predominant abundance, not harboring a central role in the network of the GIT microbiome, yielding only low-stress centrality except for few tps (BF: M08; NBF: M01, M02). This may be due to *Bifidobacterium's* unique metabolic ability to metabolize human milk oligosaccharides (HMOs), which limits its niche overlap with other microbes.

Interestingly, this was not the case in the GIT of NBF infants which did not receive breastmilk containing HMOs. In these infants, *Bifidobacterium* likely relied on more syntrophic interaction with other microbes. While in BF infants, *Bifidobacterium* mainly co-occurred with *Escherichia-Shigella* and *Enterococcus*, in NBF infants, it associated with a wider range of partners. Contrary to expectations, *Bifidobacterium* did not have a higher relative abundance in BF than in NBF infants' GIT microbiome (Fig. 5d).

Strain tracking of MAGs revealed several *Bifidobacterium* strains in several infants (Fig. 6). Even though MAGs of overall seven *Bifidobacterium* species could be detected (Fig. S18), only three of them, including *B. adolescentis*, *B. longum*, and *B. pseudocatenulatum,* were trackable in just one infant at two consecutive tps (Fig. 6). *Bifidobacterium longum*, on the other hand, could be tracked in four infants between S3 and M01 (Fig. 6).

Comparing BF and NBF infants, we observed only a few bacterial genera that were significantly differentially abundant between the two groups (Fig. S19). *Lactobacillus* was lower in relative abundance in NBF infants in months M08 to M11 (Aldex2, M08 $P = 0.013$, M09 $P = 0.004$, M10 $P = 0.006$, and M11 $P = 0.001$). *Intestinibacter* was significantly differentially abundant, showing higher relative abundance in M03, M04 and M08 in NBF infants (Aldex2, M03 $P = 0.015$, M04 $P < 0.001$, and M08 $P = 0.011$). This strong differential abundance of *Intestinibacter* at several time points is interesting, as *Intestinibacter* did not occur to be prominent in any other analysis. Interestingly, between M05 and M07, no genera were differentially abundant.

## Persistent colonization is sparse in the first year of life

A subset of samples was subjected to shotgun sequencing, resulting in the assembly of 133 high-quality MAGs (completeness >90%, contamination <5%), derived from 65 samples from 21 infants. An overview of the samples is provided in Fig. S1. Using these data, strains of several bacterial species, in addition to *Bifidobacterium,* were tracked in several infants at various tps (Fig. 6). Surprisingly, only a few strains could be detected in an infant across more than two or three consecutive tps, which would typically indicate persistent colonization of the lower GIT by that strain. Persistent colonization was only observed for few species, including *Aeromonas caviae*, three *Bifidobacterium* species, *Blautia A wexlerae*, *Faecalibacterium prausnitzii_D*, several *Streptococcus* species, for example, *Streptococcus parasanguinis_E* and *Staphylococcus* species, for example, *Staphylococcus lugdunensis*, *Rothia*, and *Veillonella.*

Also, the archaeal profile did not reveal a steady colonization. Only 134 out of 442 stool samples and 224 out of 415 oral samples gave a high-quality amplicon output (see also overview Fig. S2). Archaeal signals were only detected in S3 for infants i20 and i21, but not from M01 to M12. Archaeal presence in the lower GIT was confirmed early in life, but a highly variable and transient pattern was observed both between infants and over time (Fig. 7a and b).

*Methanobrevibacter* was the most predominant archaeal genus in the GIT similar to the oral microbiome. Some infants exhibited several archaeal genera at various tps (e.g.,

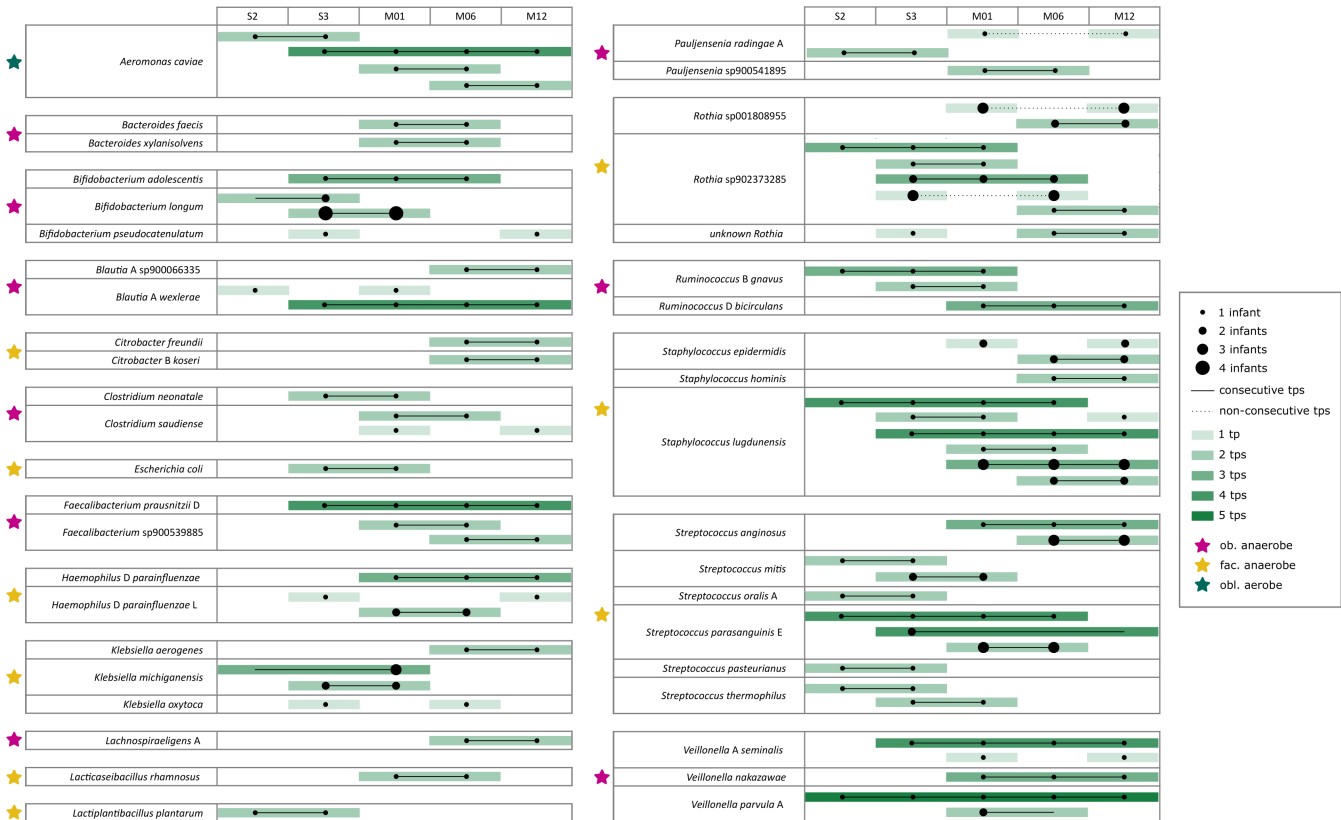

**FIG 6** Species for which strains could be tracked within one or more infants (number represented by dot size) over time (number of time points represented by the intensity of green color). Stroke style indicates if the strain could be tracked at consecutive time points (line) or non-consecutive time points (dashed). The oxygen requirement of the respective strain is highlighted by colored stars: pink: obligate anaerobes, yellow: facultative anaerobes, and petrol: obligate aerobes.

i24 and i26), while others showed only one genus at single tps (e.g., i07 and i09). Statistical analysis using Fisher's *t* test revealed no significant differences in archaeal presence/absence based on feeding type or birth mode. However, *Methanobrevibacter* and *Methanosphaera* were more common in NBF infants, whereas BF infants displayed a more diverse archaeal pattern, with higher relative abundances of unclassified Nitrosos- phaeraceae and *Candidatus* Nitrosocosmicus, possibly due to mouth-to-skin contact during breastfeeding (Fig. 7a).

Kraken/Bracken of metagenomic sequences identified five archaeal species: *Methano- brevibacter_A_sp900766745*, *Methanobrevibacter_A_smithii*, *Methanobrevibacter_A_woe- sei*, *Methanosphaera_cuniculi*, and *Methanosphaera_sp900322125*, with *Methanobrevibacter_A_sp900766745* being the most predominant (Fig. S21). All 21 infants with metagenomic data, showed archaeal signatures in their GIT across 41 samples, but at very low relative abundances (<0.07%). The highest archaeal abundances were observed at M12, indicating an increase of archaea in the first year of life. We could show that infants already have archaeal signatures in their upper and lower GIT in the first month of life, but colonization takes place late or even after M12.

## Differences between BF and NBF infants' GIT microbiomes are less pro- nounced on functional levels

Comparative, functional analyses were performed on metagenome stool samples of tps M01, M06 and M12 (Fig. 8). The very high numbers of functions that were significantly differentially abundant between the tps (DeSeq2, M01 to M06 $n = 320$ with $q < 0.05$, M06 to M12 $n = 2,218$ with $q < 0.05$, M01 to M12 $n = 2,901$ with $q < 0.05$) indicate a very high dynamic of microbial potentials in the first year of life. Microbial functions with the top

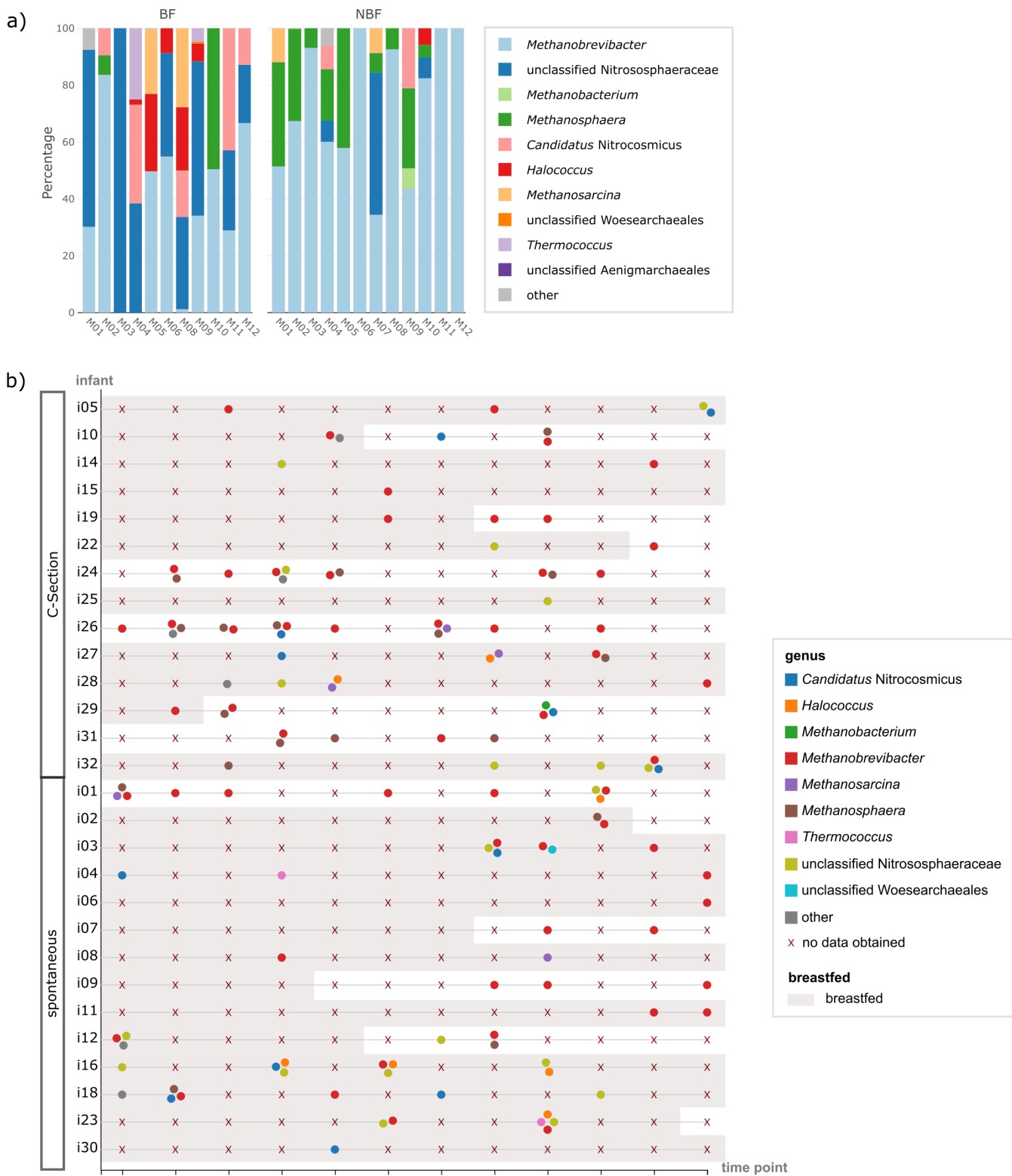

FIG 7   Panel on the GIT archaeome. (a) Stacked bar chart showing the relative abundance of the 10 most common archaeal genera in the GIT per time point (months M01 to M12); data are shown separately for breastfed (BF) and non-breastfed (NBF) infants. (b) Beeswarm plot on absence/presence of archaeal signatures in the GIT microbiome per infant and time point (months M01 to M12); time points at which the infants were BF are underlaid with gray and infants are sorted by their mode of delivery; for infants i20 and i21 no archaeal data could be obtained at any time point.

20 highest log fold-change were associated mainly with growth. This was reflected by basic metabolic pathways and a high number of genes responsible for metabolism, especially energy metabolism (e.g., oxidative phosphorylation) and carbohydrate metabolism. At M06 in comparison, genes for proteins of secretion systems involved in signaling and cellular processes were overrepresented. Examples for this were transporters and signaling proteins. The aging microbiome again shifted towards metabolism when the infants were 1 year old (M12), but with more complex pathways covering the food chain down to methane. Notably, antimicrobial resistance genes could already be found at M06 and M12.

When comparing the functional potential of the GIT microbiome of BF and NBF infants at one tp, a slighter difference was observed (DeSeq2, M01 $n = 85$ with $q < 0.05$, M06 $n = 19$ with $q < 0.05$, and M12 $n = 195$ with $q < 0.05$; Fig. S20) than it was seen between tps. Additionally, differences between BF and NBF infants' GIT microbiomes are less pronounced on functional levels than on taxonomic levels.

At M01, no gene was significantly differentially abundant for NBF infants (DeSeq2, all $q > 0.05$), meaning that all 85 genes were exclusively associated with BF infants. As most of these functions are also somehow connected with metabolism, the GIT microbiome of infants that receive BM could offer a higher number of genes that are needed to metabolize this very complex food. At M06, five genes were exclusive for NBF infants: ABC transporters and proteins for genetic information processing or signaling and cellular processes. In contrast to this, proteins of secretion systems, metabolism, and signal transduction were assigned to BF infants. In the GIT microbiome of infants of 1 year of age (M12), the nature of genes (higher hierarchical levels) that were assigned either to BF or NBF infants are very similar, even though on the lowest hierarchical level differences were observed. It can be concluded that the GIT microbiome is fulfilling the same grand functions, but with a different taxon-dependent genetic inventory.

## DISCUSSION

In infants, initial microbiome development is influenced by factors such as mode of delivery and feeding type (73). Throughout the first year of life, additional factors—including the introduction of solid foods, teething, and infants' increased mobility—shape the microbiome's structure (19, 73, 74). These changes expose infants to new microbes and create diverse environments and conditions for microbial growth, facilitating the establishment of obligate anaerobes with tense networks. We could show that birth mode impacts the initial oral and GIT colonization, but feeding has higher impacts especially later within the first year of life. Our study provides valuable insights into the early development and transition of the oral microbiome highlighting differences between BF and NBF infants. We could determine a distinct time frame in which the oral microbiome transitions most and showed that this time frame lagged between BF and NBF infants.

Breastfeeding is recognized as a significant factor influencing the GIT but also oral microbiome (75). Our results confirm that breastfeeding notably impacts microbiome composition at several time points for both oral and GIT samples. Specifically, BF infants exhibit a more defined transitional phase in their oral microbiome compared to NBF infants. This transitional phase is marked by a decrease in *Streptococcus* and the emergence of new genera such as *Granulicatella*, *Neisseria*, *Veillonella*, *Alloprevotella*, and *Leptotrichia*. It is also characterized by increased alpha-diversity of the colonizing species and significant changes in the microbial community as indicated by beta-diversity. As this transitional phase occurs earlier in NBF infants (months 1–3) than in BF infants (months 4–6), we can infer that breastfeeding supports a later, but more defined, maturation of the oral microbiome. By month 7, after the BF infants' transitional phase has ended, the microbiomes of both groups become more similar in terms of alpha- and beta-diversity, as well as differentially abundant taxa. This convergence is likely influenced by the introduction of solid food, which acts as a leveling factor between BF and NBF infants' microbiomes. This aspect was already discussed before (76, 77) but still,

**FIG 8** Hierarchical KEGG functional annotation of genes of highest top 30 log fold-change rates of pairwise comparison of (i) months M01 (orange) and M06 (green) and (ii) M06 (green) and M12 (purple); negative log fold change: red, higher in (a) M01 (vs. M06) or (b) M06 (vs. M12); positive log fold change: blue, higher in (a) M06 (vs. M01) or (b) M12 (vs. M06); significant

**Fig 8 (Continued)**

$q$-value indicated by asterisks and base mean indicated by bar charts, colored by the respective time point; abbreviation env. info. processing = environmental information processing.

complete cessation of BM rather than the introduction of solid food is the major driver for aligning the microbiomes (73).

Microbial network complexity also differs significantly between BF and NBF infants. NBF infants have more complex networks from the first month, with multiple genera exhibiting high-stress centrality and consistent network structure over time. In contrast, the microbial networks in BF infants are less complex, with fewer genera sharing similar niches. This difference in complexity is most distinct in the first 6 months and can be attributed to the differing nature of BM and formula milk. Factors listed by reference (6) include transmission of bacteria only through BM (78, 79), various milk components influencing the attachment of bacteria to the oral cavity (80) and utilization of different carbohydrates in breastmilk (e.g., HMOs) and formula milk by bacteria (81, 82). After 6 months, the microbial network of BF infants becomes more complex, likely reflecting the introduction of solid food and the subsequent increased microbial colonization and co-occurrences for improved nutrient degradation.

In the first month of life, the human skin significantly contributes to the microbial influx into the oral cavity (8, 69). This is evidenced by the high relative abundances of *Staphylococcus*, a skin- and mucosa-associated facultative anaerobe. Our data showed no significant differences in *Staphylococcus* abundance between BF and NBF infants, suggesting substantial skin-to-oral transfer independent of feeding mode. However, from month 3 onwards, *Staphylococcus* presence diminished and appeared only sporadically in the co-occurrence networks with low centrality, indicating its transient colonization in the oral cavity during early life.

In addition to the human skin, transmission from other individuals as well as environmental exposures can also be considered as the origin of the normal microbiome of the oral cavity. In fact, the predominant component of the oral microbiome, for example, *Streptococcus* is transmitted through these routes (6). This facultatively anaerobic bacterial taxon, is known for its role in carbohydrate metabolism and is in fact considered as a pioneer species in oral microbiome assembly (70). Despite its high relative abundance especially in the first 3 months (>60% in both BF and NBF infants), *Streptococcus* exhibited low network centrality, suggesting its inferior co-occurrence with other microbes and its independent functionality. Interestingly, this bacterial taxon showed a higher dominance in BF infants, as reflected by both its relative abundance and network centrality. *Streptococcus* decreases in abundance, with new microbial members emerging, marking a transitional phase in the oral microbiome.

Microbes from the oral cavity are constantly swallowed and transitioned through the GIT. Source tracking analyses showed that the contribution of the oral microbiome to the GIT microbiome was overall modest and even decreased over time. Key genera, such as *Bifidobacterium*, *Staphylococcus*, and *Streptococcus*, were identified as being transferred from the oral cavity to the GIT. The presence of these genera in both microbiomes highlights the interconnectedness of the oral and GIT microbiomes in early life. However, it is believed that the similarity of oral and GIT microbiomes decreases over time due to the development and maturation of gastric barrier, including gastric pH, motility, and enzyme production by 1 year of age (23, 83–85). Wernroth et al. (86) already highlighted that some OTUs are shared between saliva and fecal samples. They showed that one *Veillonella* OTU is mainly shared and that the similarity between saliva and stool decreased over time (86).

It should be mentioned that our methods did not allow for a distinction of living and dead microorganisms, and we cannot make definitive statements about colonization status in the early months of development, as microbial signatures might remain

detectable throughout the GIT, although the microorganisms from the oral cavity might have died during passage.

The detection of archaea in the infant GIT as early as the first month of life provides new insights into the microbial ecology of the infant GIT. *Methanobrevibacter* is the predominant archaeal genus, with its abundance increasing over time. The presence of other archaeal genera, however, particularly skin-associated ones in BF infants, indicates a more diverse archaeal community possibly influenced by close contact during breastfeeding. Although archaea are detected at low relative abundance, their presence becomes more pronounced by month 12 (M12), indicating a gradual establishment in the GIT microbiome.

Our findings support findings from other studies that infants under 1 year already carry archaea in their intestinal tracts (22, 23). Archaea found in human colostrum and BM (76) suggest vertical transfer from mother to child during breastfeeding. Other potential sources include cow milk, dairy products (77), and the archaeomes of other humans, exposing both BF and NBF infants to archaeal sources.

The sporadic presence and absence of archaeal signatures over time support the continuous transition of archaea from the environment into and through the infant's intestinal tract. More frequent detection of archaea in oral samples compared to stool samples implies that the input of archaea exceeds their successful colonization in the lower intestinal tract.

Next to strictly anaerobic archaea like *Methanobrevibacter*, also anaerobic bacteria played a massive role in the stabilization of microbiomes. *Veillonella* and *Alloprevotella* appeared in the oral cavity and GIT microbiomes, indicating early colonization by anaerobic bacteria. *Neisseria*, an obligate aerobe, plays a crucial role in facilitating the survival of anaerobes in the oral cavity by creating microenvironments with lower oxygen levels (71).

Interestingly, despite the increasing relative abundance of *Bifidobacterium* over time (which is expected due to the decreasing levels of oxygen), this bacterial taxon does not play a central role in microbial networks, especially in BF infants. This could be due to its unique metabolic niche of HMO conversion, highlighting the metabolic specialization and niche partitioning within the infant GIT microbiome. Notably, the strain tracking of *Bifidobacterium* in several infants over time, and therefore its persistence, highlights their colonizing potential and possible role in maintaining GIT health and stability. Other strains that we could track were *Blautia_A wexlerae* (obl. anaerobe) and *Faecalibacterium prausnitzii_D* (obl. anaerobe), both highly beneficial bacteria with anti-inflammatory properties (87) that are known to maintain GIT health by aiding in the production of short-chain fatty acids (SCFAs) (88). Further, positive tracking events could be observed for *Streptococcus parasanguinis_E* (obl. anaerobe) and *Veillonella_A seminalis* (obl. Anaerobe), which are both involved in early colonization in the oral cavity but can also be found in the GIT (72, 89). *Veillonella parvula_A* (obl. anaerobe), which like other *Veillonella* species, plays a role in maintaining a balanced GIT microbiome (89) could also be tracked over time. On the other hand, we see also bacterial species with not such a clear role, like *S. lugdunensis* (fac. anaerobe), a skin commensal whose presence in the GIT is less understood (90) and can cause severe infections, especially in hospital (91) and *A. caviae* (fac. anaerobe), an opportunistic pathogen that is rare in healthy infants (92). *S. lugdunensis* could be tracked even in several infants.

While the GIT microbiomes of BF and NBF infants differed in composition and complexity, the functional potential of these microbiomes is rather influenced by age than by feeding mode as indicated by the functional analysis of GIT metagenomes. Our results indicated a dynamic shift in gene abundance over time, with significant differences between months, but overall functional redundancy across BF and NBF infants.

## Conclusion

Our findings underscore the dynamic nature of the microbiome during infancy and the significant impact of breastfeeding on microbial development throughout the entire digestive tract. We could show that the oral and GIT microbiomes of BF infants undergo distinct phases of increased dynamics within the first year of life. In contrast, the microbiomes of NBF infants are more mature from the first month, leading to a steadier development without distinct transitional phases in the first year. Additionally, we found that archaeal signatures are present in infants under 1 year of age, but they do not form a stable archaeome. While the oral microbiome initially influenced the GIT microbiome during infancy, the GIT microbiome gradually stabilized and differentiated over the first year of life. This transition was marked by a decreasing influence of the oral microbiome on the GIT microbiome composition, suggesting a maturation of the GIT microbial community independent of early oral influences. These results provide valuable insights into the often-overlooked aspects shaping the infant microbiome development.

## ACKNOWLEDGMENTS

The authors thank the Medical University of Graz for the computational resources of the MedBioNode and the Life Science Compute Cluster (LiSC) operated by the Computational Systems Biology group at the University of Vienna. The authors thank the Medical University of Graz ZMF Galaxy Team: Core Facility Computational Bioanalytics, Medical University of Graz, funded by the Austrian Federal Ministry of Education, Science, and Research, Hochschulraum-Strukturmittel 2016 grant as part of BioTechMed Gral 2016 grant as part of BioTechMed Graz. The authors thank the Department of Obstetrics and Gynecology of the Medical University of Graz for the sample collection, namely Bettina Amtmann and Petra Winkler. A special thanks go to the participants of this study for providing samples and information.

This research was funded in whole or in part by the Austrian Science Fund (FWF) [Grant-DOI 10.55776/KLI784 and 10.55776/DOC31]. For open access purposes, the author has applied a CC BY public copyright license to any author accepted manuscript version arising from this submission. The study was financially supported by the City of Graz (to M.R.P. and C.N.) and the Austrian Commission for UNESCO and L'ORÉAL with the L'OREAL Fellowship for Women in Science (to M-.R.P.). R.M. and M-.R.P. was trained in the Doctoral Program MolMed, T.S. was trained in the Doctoral Program RespImmun and C.N. was trained in the PhD Program DP-iDP at the Medical University of Graz.

Conceptualization was done by the following: M.-.R.P., E-C. W., V. K.-K., E.J-.K., and C.M-.E. Methodology was done by the following: C.J.N., M.-.R.P., T.S., C.K., A.M., and C.M-.E. Formal analysis was done by the following: C.J.N. Investigation was done by the following: C.J.N., M-.R.P., and P.Y.W. Writing—original draft was done by the following: C.J.N., R.M., P.Y.W., and C.M-.E. Writing—review and editing was done by the following: C.J.N., R.M., M-.R.P., P.Y.W., T.K., V.H., T.S., P.M., A.M., C.K., E.J-.K., and C.M-.E. Visualization was done by the following: C.J.N. Supervision was done by the following: C.M-.E. and E.J-.K. Project administration was done by the following: C.J.N. and M.-.R.P. Funding acquisition was done by the following: C.J.N., M.-.R.P., C.M-.E., and E.J-.K.

No potential competing interest was reported by the author(s).

## AUTHOR AFFILIATIONS

[1]Diagnostic and Research Institute of Hygiene, Microbiology and Environmental Medicine, Medical University of Graz, Graz, Styria, Austria

[2]Department of Food Science and Nutrition, The Hong Kong Polytechnic University, Hong Kong, Hong Kong

[3]Research Institute for Future Food (RiFood), The Hong Kong Polytechnic University, Hong Kong SAR, China

[4]BBMRI-ERIC, Graz, Styria, Austria

[5]Department of Obstetrics and Gynecology, Medical University of Graz, Graz, Styria, Austria

[6]Research Unit Early Life Determinants (ELiD), Medical University of Graz, Graz, Styria, Austria

[7]BioTechMed, Graz, Styria, Austria

## AUTHOR ORCIDs

Charlotte J. Neumann http://orcid.org/0000-0003-0034-4199

Pei Yee Woh http://orcid.org/0000-0001-5950-7883

Alexander Mahnert http://orcid.org/0000-0001-7083-8894

Christina Kumpitsch https://orcid.org/0000-0002-2077-2839

Evelyn Jantscher-Krenn http://orcid.org/0000-0003-3568-891X

Christine Moissl-Eichinger http://orcid.org/0000-0001-6755-6263

## FUNDING

| Funder | Grant(s) | Author(s) |
| --- | --- | --- |
| Austrian Science Fund (FWF) | 10.55776/KLI784 and 10.55776/DOC31 | Evelyn Jantscher-Krenn |

## AUTHOR CONTRIBUTIONS

Charlotte J. Neumann, Conceptualization, Data curation, Formal analysis, Funding acquisition, Investigation, Methodology, Validation, Visualization, Writing – original draft, Writing – review and editing | Rokhsareh Mohammadzadeh, Writing – original draft, Writing – review and editing | Pei Yee Woh, Investigation, Writing – original draft, Writing – review and editing | Tanja Kobal, Writing – review and editing | Manuela-Raluca Pausan, Conceptualization, Funding acquisition, Investigation, Methodology, Project administration, Writing – review and editing | Tejus Shinde, Methodology, Writing – review and editing | Victoria Haid, Formal analysis, Writing – review and editing | Polona Mertelj, Methodology, Writing – review and editing | Eva-Christine Weiss, Conceptualization | Alexander Mahnert, Methodology, Software, Writing – review and editing | Christina Kumpitsch, Methodology, Writing – review and editing | Evelyn Jantscher-Krenn, Conceptualization, Funding acquisition, Methodology, Project administration, Resources, Supervision, Writing – review and editing | Christine Moissl-Eichinger, Conceptualization, Funding acquisition, Methodology, Project administration, Resources, Supervision, Writing – original draft, Writing – review and editing.

## DATA AVAILABILITY

Data, tables, and scripts that support our findings are openly available in our GitHub Repository at https://github.com/CharlotteJNeumann/InfantDevelopmentTRAMIC. The generated 16S rRNA gene amplicon data are accessible in the European Nucleotide Archive under the study accession number PRJEB77729.

## ETHICS APPROVAL

This study has been registered on clinicaltrials.gov (NCT04140747). The samples were collected under the ethical approval number 28-524 ex15/16 by the respective local ethics committees, the Ethics Committee at the Medical University of Graz, Graz, Austria and in adherence to the principles outlined in the Declaration of Helsinki.

## ADDITIONAL FILES

The following material is available online.

## Supplemental Material

**Supplemental Figures (mSystems01071-24-s0001.pdf).** Figures S1 to S21.

## Open Peer Review

**PEER REVIEW HISTORY (review-history.pdf).** An accounting of the reviewer comments and feedback.

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
