## [Reviewer comments · mSystems]

First-Year Dynamics of the Anaerobic Microbiome and Archaeome in Infants' Oral and Gastrointestinal Systems

Charlotte Neumann, Rokhsareh Mohammadzadeh, Pei Yee Woh, Tanj Kobal, Manuela Pausan, Tejus Shinde, Victoria Haid, Polona Mertelj, Alexander Mahnert, Christina Kumpitsch, Evelyn Jantscher-Krenn, and Christine Moissl-Eichinger

Corresponding Author(s): Christine Moissl-Eichinger, Medizinische Universitat Graz

Review Timeline:

Submission Date:	August 8, 2024
Editorial Decision:	September 25, 2024
Revision Received:	October 23, 2024
Accepted:	November 26, 2024

Editor: Daniel Garrido

Reviewer(s): Disclosure of reviewer identity is with reference to reviewer comments included in decision letter(s). The following individuals involved in review of your submission have agreed to reveal their identity: Elaine M Haase (Reviewer #3); Justin Shaffer (Reviewer #4)

Transaction Report:

DOI: <https://doi.org/10.1128/msystems.01071-24>

Re: mSystems01071-24 (First-Year Dynamics of the Anaerobic Microbiome and Archaeome in Infants' Oral and Gastrointestinal Systems)

Dear Dr. Christine Moissl-Eichinger:

Revision Guidelines

Sincerely,
Daniel Garrido
Editor
mSystems

Reviewer #1 (Comments for the Author):

This is an interesting paper that examines the oral and fecal microbiomes in the first year of life for a cohort of infants. Although the cohort is moderately sized, the number of samples are impressive and the findings appear intriguing, I felt that there could have been a bit more explanation of the approaches and discussion of possible limitations.

A major question being addressed is the difference between breast-fed and non-breast-fed babies. I was not quite sure though what exactly those groups are, maybe I missed that? Is it by sample and whether they were breastfed that month? So in some

cases there are comparisons done between samples from the same individual? Or is there another way that the groups were divided by subject? It would be good to know and I could not easily tell.

Additionally there are variables that could confound analyses, for instance antibiotics, solid food introduction, dental hygiene practises, tooth eruption, etc. The authors do provide a table showing the distribution of some of these with regard to delivery mode but do not show their distribution with regard to breastfeeding. Since breastfeeding is such a focus it would be good to see a table with that. I also note that it is possible to use distance based RDA to examine the effect of multiple variables on bacterial populations and maybe that would be a possible approach to examine possible confounders.

I also had a few minor comments, as follows:

Throughout, GIT microbiome is used while it would be more accurate to say fecal microbiome.

Line 58-59, "one of the most intricate one" is not grammatically correct.

Line 209-210 "ASVs with < 1 read" doesn't make sense, should be = I guess.

Line 536-537 "The GIT microbiome develops steadier" is grammatically incorrect, should use the adverb version "more steadily"

Line 674 "increased mobility" It was a bit unclear to me at least if this refers to increased mobility of the infant or the microbes.

Line 762: There is a missing reference "(XXX)" - note that it wouldn't bother me if a reference was not included and this was stated as a hypothesis

References 74 and 79 are the same.

Reviewer #2 (Comments for the Author):

The presented manuscript by Neumann et al. has revealed changes of oral and gut microbiome in early in life, and found relationship and distinct maturation between them, in addition to whether breast-fed or not.

The findings are very interesting with enough supportive data. This reviewer has one point to ask regarding the number of bacteria (i.e., absolute composition) in early in life, as the authors showed relative composition and functionality of oral and gut microbiome. The changes in number of bacterial colonization in infants may affect the overall phenotype related to microbiota.

Reviewer #3 (Comments for the Author):

Information regarding recruitment and informed consent is referred to in a previous publication. This should be reiterated for this manuscript.

This is an interesting longitudinal study describing the dynamics of infant oral and stool microbiomes comparing those from breast-fed and non-breast fed infants. Comprehensive and n-depth 16S rRNA gene analysis was performed. Data analysis indicated that BF babies microbiomes had a transition period until mature microbiomes appeared as compared to NBF babies whose microbiomes were more heterogenous from the beginning and lacked any definitive transition period. Special emphasis was given to changes in the anaerobic microbiome and archaeome.

There are a few areas that need clarification as indicated below:

Line 149. What was the protocol for obtaining oral swabs? How many surfaces sampled, buccal mucosa, tongue, gums?

Line 152. How were the stool samples obtained? Anal swab? Sample off of a diaper?

Lines 159 and 160. Source of lysozyme and mutanolysin.

Line 167. What is Inhibitex and what is the source?

Line 178. As controls, were mock communities incorporated for both the oral and stool protocols?

Line 278. Why could metagenomic data be obtained for only 9 oral samples?

Was there any influence of mode of delivery the initial oral and GIT colonization? The data is available and there should be a little more discussion of this.

Line 465. Define BM at first occurrence.

Line 762. Please fill in (XXX).

Reviewer #4 (Comments for the Author):

general comments:

This paper describes how breastfeeding impacts microbial composition at several time points for both oral and gastrointestinal microbiota. Breastfed infants have a more defined transitional phase in their oral microbiome compared to non-breastfed infants. My major suggestion is that the English be checked as there are several mistakes throughout (see my comments below for examples).

- if possible avoid using abbreviations and use the full term: GIT, BF, NBF

abstract

- line 31: I suggest to hyphenate alpha-diversity and beta-diversity

- line 39: I suggest to add an apostrophe to infants

importance

- very succinct summary - nice work!

introduction

- line 73: I suggest to choose between 'niche' and 'habitats' as there is a distinct difference in ecology - based on the list that follows I would say you are describing habitats

- line 83: You first define GIT as gastrointestinal, but here it seems you want to say gastrointestinal tract? I suggest to avoid using abbreviations altogether, in part due to this issue

- line 87: I suggest to add 'the' before 'GIT'

- line 88: I suggest to pluralize microbiome somehow - microbiomes or microbiota, and to remove 'the' before 'GIT'

- line 89: Do you mean 'interaction between the oral and GIT microbiomes'? Please rephrase to clarify

- line 90: Again, I think you mean gastrointestinal tract? You define GIT as just gastrointestinal. I see you continue to use GIT for both terms throughout, so I suggest to either not use the abbreviation or to define gastrointestinal tract as GITT

- line 97: Earlier you abbreviated species as spp. - please normalize for consistency

- line 108: Yes! I agree. Good statement

Methods

- line 140: I suggest to add a README file to your GitHub page that clarifies what the contents contains

- lines 173-174: I suggest to rephrase this sentence as it is overcomplicated:

Subsequent to the mechanical lysis process, the samples underwent a 5-minute incubation at 70{degree sign}C, succeeded by a centrifugation step at 10,000 x g for 3 minutes to segregate the beads from the supernatant

Perhaps something like this to simplify things:

After mechanical lysis, the samples were incubated at 70{degree sign}C for 5 minutes and then centrifuged for 10,000 x g for 3 minutes to separate the beads from the supernatant

- lines 181-189: Although this paragraph is clear, it is also overcomplicated and could be clarified further by rephrasing to avoid words like acquired, employed, discern, and executed.

- line 222: I suggest to hyphenate alpha-diversity (throughout)

- line 226: I suggest to hyphenate beta-diversity (throughout)

- line 231: Yay, bacdiv!

- line 348: Move 'also' to after 'could'

- line 479: I suggest to change tps to timepoints (and tp to timepoint, throughout), for clarity

- line 503: Change was to were

- line 542: Change 'lower' to 'smaller'

- line 557: I suggest to change 'were co-occurring' to 'were found to co-occur'

- line 580: add 'on' between 'relied' and 'more'

- line 601: Change was to were

- lines 700-702: Can you elaborate further on what differs between milk and formula that would cause this difference? The following support from Xiao et al. 2020 seems to only discuss features of milk - what about formula?

- line 738: good point

Reviewers Comments and Author Responses

We appreciate the time of both reviewers and are thankful for their helpful comments and valid corrections.

Please find our responses below.

Reviewer #1:

This is an interesting paper that examines the oral and fecal microbiomes in the first year of life for a cohort of infants. Although the cohort is moderately sized, the number of samples are impressive and the findings appear intriguing, I felt that there could have been a bit more explanation of the approaches and discussion of possible limitations.

Major comments

A major question being addressed is the difference between breast-fed and non-breast-fed babies. I was not quite sure though what exactly those groups are, maybe I missed that? Is it by sample and whether they were breastfed that month? So in some cases there are comparisons done between samples from the same individual? Or is there another way that the groups were divided by subject? It would be good to know and I could not easily tell.

Indeed, the distinction of breast-fed and non-breastfed infants was performed individually for each single time-point. This is now stated more clearly in lines 348 to 352. All our comparisons were based on months, and so only one sample per individual per month is included.

Additionally, there are variables that could confound analyses, for instance antibiotics, solid food introduction, dental hygiene practises, tooth eruption, etc. The authors do provide a table showing the distribution of some of these with regard to delivery mode but do not show their distribution with regard to breastfeeding. Since breastfeeding is such a focus it would be good to see a table with that. I also note that it is possible to use distance based RDA to examine the effect of multiple variables on bacterial populations and maybe that would be a possible approach to examine possible confounders.

Thank you for your comment. We included a table (Table 4) examining the distribution of the respective confounders with regard to feeding (less and longer than 6 months). We have included the statistical values associated with the different factors. Instead of RDA we have employed PERMANOVA to assess the effect of these factors on the microbiomes. In our dataset we found no strong associations beyond the feeding regimen

and birth mode (for M01 only). This is why we concentrated on the impact of the feeding regiment. See lines 353 to 360.

Minor comments

Throughout, GIT microbiome is used while it would be more accurate to say fecal microbiome.

You are right and we agree. We have adapted the term to common usage in other manuscripts, but we are happy to change it according to your suggestion if the editor advises us to do so.

Line 58-59, "one of the most intricate one" is not grammatically correct.

We changed it accordingly, thanks.

Line 209-210 "ASVs with < 1 read" doesn't make sense, should be = I guess.

We changed it accordingly, thanks. As with and after decontam, samples were excluded (including PCs and NCs), there were as a consequence some ASVs left that had 0 reads, as the samples that had some reads of this ASV were excluded. Those ASVs need to be excluded as well, so we changed it to " \leq ".

Line 536-537 "The GIT microbiome develops steadier" is grammatically incorrect, should use the adverb version "more steadily"

We changed it accordingly, thanks.

Line 674 "increased mobility" It was a bit unclear to me at least if this refers to increased mobility of the infant or the microbes.

We changed it to "infants' increased mobility" and hope that it is clear now, thanks.

Line 762: There is a missing reference "(XXX)" - note that it wouldn't bother me if a reference was not included and this was stated as a hypothesis

We changed it accordingly, thanks.

References 74 and 79 are the same.

We changed it accordingly, thanks.

Reviewer #2

The presented manuscript by Neumann et al. has revealed changes of oral and gut microbiome in early in life, and found relationship and distinct maturation between them, in addition to whether breast-fed or not.

The findings are very interesting with enough supportive data.

This reviewer has one point to ask regarding the number of bacteria (i.e., absolute composition) in early in life, as the authors showed relative composition and functionality of oral and gut microbiome. The changes in number of bacterial colonization in infants may affect the overall phenotype related to microbiota.

Thank you for your comment. We did not directly analyze the number of bacteria by any quantitative method like e.g. qPCR, so we cannot draw any conclusions on that. Still at least for stool samples, metagenomic data can somewhat be regarded as absolute abundances.

Reviewer #3

Information regarding recruitment and informed consent is referred to in a previous publication. This should be reiterated for this manuscript.

We already provided a reiterated listing of the major recruitment information in lines 138-143. Still, we rephrased to make it more clear and added some more information.

This is an interesting longitudinal study describing the dynamics of infant oral and stool microbiomes comparing those from breast-fed and non-breast fed infants. Comprehensive and n-depth 16S rRNA gene analysis was performed. Data analysis indicated that BF babies microbiomes had a transition period until mature microbiomes appeared as compared to NBF babies whose microbiomes were more heterogenous from the beginning and lacked any definitive transition period. Special emphasis was given to changes in the anaerobic microbiome and archaeome.

Line 149. What was the protocol for obtaining oral swabs? How many surfaces sampled, buccal mucosa, tongue, gums?

We added the information, thanks.

Line 152. How were the stool samples obtained? Anal swab? Sample off of a diaper?

We added the information, thanks.

Lines 159 and 160. Source of lysozyme and mutanolysin.

We added the information, thanks.

Line 167. What is Inhibitex and what is the source?

We changed it, it was Lysis Buffer as well, thanks.

Line 178. As controls, were mock communities incorporated for both the oral and stool

protocols?

Yes, we also included positive controls. We added this information here additionally, thanks.

Line 278. Why could metagenomic data be obtained for only 9 oral samples?

Unfortunately, oral samples are more difficult to sequence than stool samples e.g. as there is a high contamination of host DNA lowering the quality of microbial signals. We added this information: "...due to the challenging nature of buccal mucosa samples, such as high presence of host DNA contamination".

Was there any influence of mode of delivery the initial oral and GIT colonization? The data is available and there should be a little more discussion of this.

We included more information now in the results section (lines 353-356) and discussion (lines 694-695).

Line 465. Define BM at first occurrence.

We changed it accordingly, thanks.

Line 762. Please fill in (XXX).

We changed it accordingly, thanks.

Reviewer #4

general comments:

This paper describes how breastfeeding impacts microbial composition at several time points for both oral and gastrointestinal microbiota. Breastfed infants have a more defined transitional phase in their oral microbiome compared to non-breastfed infants.

My major suggestion is that the English be checked as there are several mistakes throughout (see my comments below for examples).

We implemented your comments and performed an additional language check on the manuscript.

if possible avoid using abbreviations and use the full term: GIT, BF, NBF

Thank you for your comment. We are convinced that abbreviations make the manuscript more compact and easily readable. Therefore, we would like to leave the decision about a general avoidance of those frequently used abbreviations like GIT, BF, NBF, tps to the editor and/or the type setting of the journal.

abstract

line 31: I suggest to hyphenate alpha-diversity and beta-diversity

We changed it accordingly, thanks.

line 39: I suggest to add an apostrophe to infants

We changed it accordingly, thanks.

importance

very succinct summary - nice work!

Thank you!

introduction

line 73: I suggest to choose between 'niche' and 'habitats' as there is a distinct difference in ecology - based on the list that follows I would say you are describing habitats

We changed it accordingly, thanks.

line 83: You first define GIT as gastrointestinal, but here it seems you want to say gastrointestinal tract? I suggest to avoid using abbreviations altogether, in part due to this issue

Introducing the abbreviation GIT for 'gastrointestinal' was a mistake from our side, GIT stands for gastrointestinal tract. We changed this in the Abstract.

We are convinced that abbreviations make the manuscript more compact and easily readable. Therefore, we would like to leave the decision about a general avoidance of those frequently used abbreviations like GIT, BF, NBF, tps to the editor and/or the type setting of the journal.

line 87: I suggest to add 'the' before 'GIT'

We changed it accordingly, thanks.

line 88: I suggest to pluralize microbiome somehow - microbiomes or microbiota, and to remove 'the' before 'GIT'

We changed it accordingly, thanks.

line 89: Do you mean 'interaction between the oral and GIT microbiomes'? Please rephrase to clarify

We rephrased the sentence to clarify that little is known about possible interaction and parallel development of the GIT and oral microbiomes.

line 90: Again, I think you mean gastrointestinal tract? You define GIT as just gastrointestinal. I see you continue to use GIT for both terms throughout, so I suggest to either not use the abbreviation or to define gastrointestinal tract as GITT

Introducing the abbreviation GIT for 'gastrointestinal' was a mistake from our side, GIT stands for gastrointestinal tract. We changed this in the Abstract.

line 97: Earlier you abbreviated species as spp. - please normalize for consistency

We changed it accordingly, thanks.

line 108: Yes! I agree. Good statement

Thanks!

Methods

line 140: I suggest to add a README file to your GitHub page that clarifies what the contents contains

Thanks for the input, we included a README file.

lines 173-174: I suggest to rephrase this sentence as it is overcomplicated:
Subsequent to the mechanical lysis process, the samples underwent a 5-minute incubation at 70{degree sign}C, succeeded by a centrifugation step at 10,000 x g for 3 minutes to segregate the beads from the supernatant

Perhaps something like this to simplify things:

After mechanical lysis, the samples were incubated at 70{degree sign}C for 5 minutes and then centrifuged for 10,000 x g for 3 minutes to separate the beads from the supernatant

We changed it accordingly, thanks.

lines 181-189: Although this paragraph is clear, it is also overcomplicated and could be clarified further by rephrasing to avoid words like acquired, employed, discern, and executed.

We changed it accordingly, thanks.

line 222: I suggest to hyphenate alpha-diversity (throughout)

We changed it accordingly, thanks.

line 226: I suggest to hyphenate beta-diversity (throughout)

We changed it accordingly, thanks.

line 231: Yay, bacdiv!

line 348: Move 'also' to after 'could'

We changed it accordingly, thanks.

line 479: I suggest to change tps to timepoints (and tp to timepoint, throughout), for clarity

Thank you for your comment. We are convinced that abbreviations make the manuscript more compact and easily readable. Therefore, we would like to leave the decision about a general avoidance of those frequently used abbreviations like GIT, BF, NBF, tps to the editor and/or the type setting of the journal.

line 503: Change was to were

We changed it accordingly, thanks.

line 542: Change 'lower' to 'smaller'

We changed it accordingly, thanks.

line 557: I suggest to change 'were co-occurring' to 'were found to co-occur'

We changed it accordingly, thanks.

line 580: add 'on' between 'relied' and 'more'

We changed it accordingly, thanks.

line 601: Change was to were

As the subject is "A subset" it requires a singular verb even though samples is plural, right?

lines 700-702: Can you elaborate further on what differs between milk and formula that would cause this difference? The following support from Xiao et al. 2020 seems to only discuss features of milk - what about formula?

The factors listed by Xiao et al. 2020 directly discuss differences between formula milk and breast milk that impact the oral microbiome. We might not have phrased it clearly enough. We hope that it is now clearer that those factors are not unique to breast milk, but different between the two milk types.

line 738: good point

Thank you

Re: mSystems01071-24R1 (First-Year Dynamics of the Anaerobic Microbiome and Archaeome in Infants' Oral and Gastrointestinal Systems)

Dear Dr. Christine Moissl-Eichinger:

Your manuscript has been accepted, and I am forwarding it to the ASM production staff for publication. Your paper will first be checked to make sure all elements meet the technical requirements. ASM staff will contact you if anything needs to be revised before copyediting and production can begin. Otherwise, you will be notified when your proofs are ready to be viewed.

Sincerely,
Daniel Garrido

Editor
mSystems

Reviewer #1 (Comments for the Author):

I am the previous reviewer 1 and the authors have acceptably responded to my critique.

Reviewer #2 (Comments for the Author):

This reviewer has no further comments.

Reviewer #3 (Comments for the Author):

The authors have satisfactorily answered all of the reviewers comments.

Reviewer #4 (Comments for the Author):

This is a great paper. It is clear that the comments and suggestions from the first round of reviews were considered. The paper is much improved.

One comment for the into:

- line 102: bifid shunt "pathway"? for clarity?